# DNA origami signal amplification in lateral flow immunoassays

Heini Ijäs ✉, Julian Trommler, Linh Nguyen, Stefan van Rest, Philipp C. Nickels, Tim Liedl & Maximilian J. Urban ✉

Lateral flow immunoassays (LFIAs) enable a rapid detection of analytes in a simple, paper-based test format. Despite their multiple advantages, such as low cost and ease of use, their low sensitivity compared to laboratory-based testing limits their use in e.g. many critical point-of-care applications. Here, we present a DNA origami-based signal amplification technology for LFIAs. DNA origami is used as a molecularly precise adapter to connect detection antibodies to tailored numbers of signal-generating labels. As a proof of concept, we apply the DNA origami signal amplification in a sandwich-based LFIA for the detection of cardiac troponin I (cTnI) in human serum. We show a 55-fold improvement of the assay sensitivity with 40-nm gold nanoparticle labels and an adjustable signal amplification of up to 125-fold with fluorescent dyes. The technology is compatible with a wide range of existing analytes, labels, and sample matrices, and presents a modular approach for improving the sensitivity and reliability of lateral flow testing.

Lateral flow immunoassays (LFIAs) are among the most widely used diagnostic tests worldwide. The scalable production, ease of use, and the possibility for rapid, instrument-free readout have made them a prime diagnostic tool in the medical field, particularly in the point of care (PoC)[1,2].

While LFIAs can provide results within minutes and with minimal effort, they are often limited by low sensitivity—i.e., detection limits ranging from micromolar down to 10–100 picomolar concentrations[3]—and thus, the initial test result in the PoC needs to be confirmed with additional laboratory-based tests. Standard laboratory-based immunoassays reach femto- to picomolar detection limits by using powerful signal amplification methods, such as enzymes (enzyme-linked immunosorbent assay; ELISA) and electrochemiluminescence (ECL), but they are time-consuming and require specialized equipment, stringent control over reaction conditions, and operation by trained personnel[3,4].

Various approaches to enhance the sensitivity of LFIAs have been demonstrated before[5]. A positive test result on commercial LFIAs is typically indicated by a colored line formed at the test zone by optical labels, such as colloidal gold, with multiple detection antibodies conjugated to a single label[2,6]. As the intensity of the colorimetric readout

scales with the number of labels, the low number of labels per captured analyte limits the sensitivity of the test. Novel, brighter labels can help to overcome this limitation by introducing a higher signal intensity per label through, e.g., fluorescence, surface-enhanced Raman scattering (SERS), or magnetic readout[7–10]. The drawback of the improved signal-to-noise ratio is that the test result has to be interpreted using label-specific readers. The increased costs of both labels and instrumentation can limit the applicability of the tests, particularly in settings such as low-resource environments and home testing.

On the other hand, various signal amplification strategies can be used to increase the number of labels per analyte at the test zone, and thus enhance the sensitivity of LFIAs without changing the type of reporter[5]. Nanostructures with repetitive motifs have been applied to link antibodies to multiple labels. These polymerization approaches include the use of DNA dendrimers and branched DNA structures[11,12], hybridization chain reaction[13,14], and DNA-tile assembly[15]. Gold labels can also be directed to aggregate at the test zone through specific surface functionalization[16,17]. However, these methods provide only limited control over the size of the applied nanostructures and over the stoichiometry of antibodies and labels. Issues arising from polydispersity, amplification of unspecific signals, and uncontrolled

Faculty of Physics and Center for NanoScience (CeNS), Ludwig-Maximilians-Universität München, Geschwister-Scholl-Platz 1, Munich, Germany. ✉e-mail: heini.ijaes@physik.lmu.de; urban.max@physik.lmu.de

aggregation of labels have so far hindered the integration of these approaches in commercial LFIA products.

DNA self-assembly allows the creation of nanostructures with Ångström precision on the scale of 10–1000 nanometers[18]. In particular, DNA origami[19] is a robust molecular programming technique that has been used to arrange molecular binders including antibodies[20–26] and a wide variety of labels[27,28]. Despite groundbreaking research work over almost two decades, DNA origami nanostructures have not yet been applied in commercial products aimed at improving human health or quality of life. Nonetheless, DNA nanotechnology enables precise control over the stoichiometry, position, and orientation of both antibodies and labels. In the scope of immunoassays, including LFIAs, this precision allows us to overcome previous limitations of LFIA signal amplification methods.

In this context, we focus on cardiac troponin I (cTnI) and human neurofilament light chain (NfL), which are two clinically significant biomarkers with distinct diagnostic applications. NfL, a marker for acute neuro-axonal injuries and neurodegenerative diseases, is only beginning to be utilized in LFIA testing[29,30]. On the other hand, cTnI is a key biomarker for cardiovascular diseases, playing a crucial role in the urgent differential diagnosis of heart attacks, particularly in non-ST elevated acute coronary syndrome[31,32]. Current high-sensitivity cTnI laboratory assays can, by definition, detect cTnI concentrations in 50–100% of healthy individuals, with typical cutoff values for elevated levels defined as 10–14 ng/L[32]. The available PoC tests typically fall far below this sensitivity, and a significant medical need for rapid (under 20 min) and sensitive (below 50 ng/L) cTnI tests remains in the PoC setting. LFIAs are globally utilized in cTnI testing, especially in settings lacking access to central laboratory facilities. With the introduction of novel labels, such as upconverting nanoparticles or superparamagnetic nanobeads, LFIA tests for cTnI have been shown to reach LoDs below 50 ng/L, at the cost of requiring fluorescence or magnetic assay readers[33–35]. Consequently, cTnI remains an essential benchmark analyte for assay development with persistent sensitivity challenges in PoC diagnostics.

Here, we present a DNA origami-based signal amplification platform for lateral flow tests, specifically for LFIAs (Fig. 1). This platform utilizes a DNA origami adapter for connecting detection antibodies to labels. Depending on the label type and DNA origami design, the number of labels per detection antibody can range from one to several hundred, offering adjustable amplification factors to meet the desired sensitivity for signal-amplified LFIAs. In our experiments, we employ sandwich immunoassays with monoclonal antibodies for the detection of cTnI and NfL. This method is adaptable to other analytes through the selection of appropriate detection molecules. Additionally, our approach can be integrated with the full range of LFIA labels, from industry-standard colorimetric gold labels to the variety of novel labels that have been developed in recent years. In the current study, we

utilize fluorescent dyes and 40-nm gold nanoparticles. Through its compatibility with diverse analytes, detection molecules, and labels, our amplification technology is designed for seamless integration into existing LFIA products and mass production workflows.

## Results

### The DNA origami signal amplification structure

The DNA origami signal amplification structure functions as an adapter that connects detection antibodies with labels in a tunable stoichiometry. Here, we use a common six-helix bundle (6HB) design, which comprises six parallel, interconnected DNA duplexes, and has a length of 490 nm and a diameter of 8 nm in solution[36]. The main functional features necessary for signal amplification are the two binding domains at the ends of the 6HB and a signal amplification domain in between (Fig. 2a).

The binding domains feature up to 12 single-stranded DNA (ssDNA) staple overhangs for either direct hybridization with oligonucleotide analytes or for attachment of DNA-conjugated recognition molecules, such as antibodies. In all experiments presented in this work, each binding domain has three overhangs, resulting in a total of six overhangs per 6HB. After an initial HPLC purification step to remove excess staple strands, the 6HBs were used for detection of DNA analytes without further modification. For the detection of cTnI, the binding domains were functionalized with monoclonal anti-cTnI IgG antibodies via DNA hybridization. The detection antibodies were conjugated to DNA handles using non-site-specific labeling of lysine residues (Fig. S1). The 6HBs were then incubated with a 6-fold molar excess of antibodies to form Ab-6HB conjugates. The conjugates were employed in LFIAs without further purification. After characterizing the conjugates with agarose gel electrophoresis (AGE) (Fig. S2) and extracting them from the leading band, transmission electron microscopy (TEM) imaging revealed conjugated antibodies bound to the binding domains at the ends of the 6HBs (Fig. 2b and Figs. S3–S4). Typically, 1–2 antibodies per binding domain could be observed.

The amplification domain of the 6HB contains 180 ssDNA overhangs designed for binding DNA-conjugated labels. To demonstrate the signal amplification strategy, we used two different labels: DNA-functionalized 40-nm gold nanoparticles (gold-DNA) and Alexa Fluor 647-labeled DNA oligonucleotides (A647-DNA). The colorimetric signal of the gold-DNA labels is visible to the naked eye, while the fluorescence from A647-DNA can be detected using a fluorescence reader.

To study the attachment of gold-DNA labels to the 6HBs, the 6HBs were incubated with an excess of 20-nm gold-DNA labels, and subsequently gel purified. TEM analysis of the assemblies shows that the binding sites on the 6HB can facilitate a dense binding of labels along the entire length of the amplification domain (Fig. 2c, left panel). TEM images of the 6HBs complexed with 40-nm gold-DNA labels are shown in Fig. S5. The attachment of the A647-DNA labels to the 6HBs was studied by means of AGE. The presence of colocalized fluorescence signals from both the SYBR Safe DNA stain and the A647 confirms the hybridization of A647-DNA with 6HBs (Fig. 2c, right panel). Throughout this study, we used a 6HB featuring 3 ssDNA overhangs at each binding domain and 180 ssDNA overhangs at the amplification domain. The amplification factor was tuned by adjusting the molar ratio of A647-DNA or gold-DNA labels relative to the 6HB during the conjugation process (Fig. 2c and Fig. S6).

Signal amplification is achieved by incorporating the DNA origami structure into a conventional LFIA test strip comprising a nitrocellulose membrane striped with capture reagents—e.g., antibodies or streptavidin—as well as pads for sample application, buffer application, storage of dry reagents, and liquid absorption after the nitrocellulose membrane (Fig. 2d). For running assays with A647-DNA labels, the Ab-6HB conjugates were preassembled with the desired excess of A647 labels prior to the assay. When utilizing gold-DNA, the labels were dried on a pad upstream of the sample

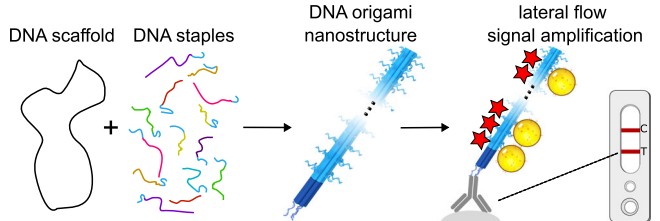

**Fig. 1 | Lateral flow immunoassay signal amplification with DNA origami.** Structurally precise DNA nanostructures are prepared with the DNA origami self-assembly method. The DNA origami is applied as an adapter that connects a specified, low number of detection antibodies to multiple labels. The greatly enhanced number of labels per antibody and per bound analyte at the test line leads to a stronger signal and makes lower analyte concentrations detectable by the naked eye.

application zone. The LFIA tests were then run in two steps (Fig. 2e). First, the liquid sample was mixed with the Ab-6HB conjugates and applied on the sample application pad. In the second step, running buffer was applied to release the dried gold-DNA labels. This induces a capillary flow that carries all components to the test and control lines, where they can interact with the capture reagents. An intense colorimetric signal is generated when a large number of gold-DNA labels bind to the amplification domains of the 6HBs. Drying of the particles is not only imperative for the long-term stability of the gold labels and for maintaining a simple user protocol without additional label handling, but it also leads to an improved test performance through a slower release of the gold-DNA labels. We observed that this prevents unwanted early interactions that can cause label-mediated crosslinking, sedimentation of high-molecular-weight 6HB-gold assemblies, and signal loss at the test line.

### Controlling the amplification factor

To show that the DNA origami allows for precise control over the signal amplification factor on lateral flow tests, we used the 6HBs with A647-DNA labels for detecting a biotinylated DNA oligonucleotide. By selecting biotin-DNA as the model analyte, the amplification system can be studied without limitations arising from, e.g., potential slow or low-affinity antibody-antigen binding reactions. The biotin-DNA can be detected without antibodies through two fast (high forward rate constant, $k_{on}$) and high affinity (low dissociation constant, $K_d$) binding reactions: biotin-streptavidin interaction ($K_d = ~10^{-15}$ mol/L) and the hybridization of a 26 base pairs long DNA double strand. The illustration in the left panel of Fig. 3a shows the molecular schematic of the structure that forms at the test line in the assay. When a sample containing the complementary biotin-DNA analyte is mixed with the A647-

labeled 6HBs, the analyte is bound through DNA hybridization at the binding domains. The 6HB-bound analytes are then captured at a streptavidin test line.

Prior to running the biotin-DNA fluorescence assays, 6HBs were mixed with either 50, 100, or 200 A647-DNA labels per 6HB ($n = 50$, 100, or 200) to adjust the amplification factor. Here, $n$ thus refers specifically to the molar ratio of the labels and the 6HBs, while the number of handles for fluorophore binding in our design is only 180. The highest excess, $n = 200$, was applied to ensure that all available binding sites were saturated. A small volume (5 µL) of biotin-DNA in milli-Q H$_2$O was mixed with the 6HB-A647, added to the test strip, and flushed over the nitrocellulose membrane with running buffer. As an unamplified reference ($n = 1$; bottom row of Fig. 3a), the biotin-DNA analyte was mixed with a complementary, A647-labeled DNA oligonucleotide instead of 6HB-A647. The concentration of the direct labeling probe was the same as the concentration of A647-DNA labels in the $n = 100$ sample.

The test line fluorescence images presented in Fig. 3a and the fluorescence intensity values plotted against the biotin-DNA analyte concentration in Fig. 3b show that the test line signal can be both amplified and controlled with the 6HBs. In the presented experiment, only 6.25 nmol/L or higher concentrations of biotin-DNA were detectable with direct labeling ($n = 1$). At smaller concentrations, the number of fluorophores at the test line is below the threshold that can be distinguished from the background fluorescence. 6HBs increase the number of fluorophores at the test line, and thus the sensitivity of the assay increases with $n$. With $n = 200$, biotin-DNA concentrations down to 50 pmol/L could be detected, corresponding to a sensitivity improvement by two orders of magnitude (125-fold). When taking into account that the number of accessible DNA extensions at a given site of a DNA origami structure ranges from *ca.* 70 to 90%[37] and that dye

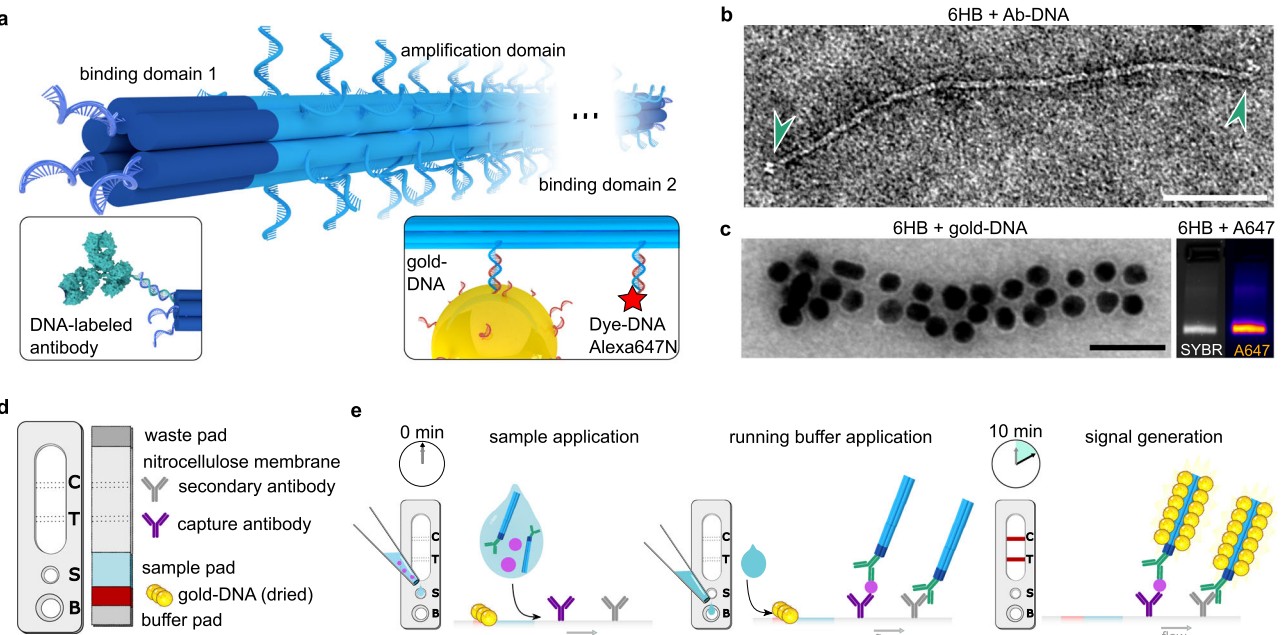

**Fig. 2 | DNA origami and LFIA test design. a** A schematic presentation of the design of a DNA origami 6HB with an adjustable number of binding domains (dark blue) and one amplification domain (light blue). For LFIAs, antibodies are attached to the binding domains. DNA-conjugated labels, such as gold nanoparticles or fluorophores, bind to the amplification domain. **b** Transmission electron microscopy (TEM) image of a detection antibody-6HB conjugate (Ab-6HB) after incubation of 6HBs with a 6-fold molar excess of DNA-conjugated detection antibodies (Ab-DNA) and gel purification. The bound antibodies are indicated by arrows (scale: 100 nm). **c** Decoration of the 6HBs with labels. The TEM image in the left panel

shows a 6HB after incubation with DNA-labeled gold nanoparticles; here, 20-nm particles for the purpose of demonstration (scale: 100 nm). The right panel shows an agarose gel electrophoresis (AGE) analysis of SYBR Safe DNA stain and Alexa Fluor 647 (A647) fluorescence of 6HBs after incubation with a 200-fold molar excess of A647-labeled oligonucleotides. **d** A general design of a LFIA test used for the integration of DNA origami signal amplification. **e** LFIA running protocol. The amplified signal is generated when the sample-DNA origami mixture interacts with both the labels and the immobilized antibodies on the test (T) and control (C) lines. Source data for panels b and c are provided as a Source Data file.

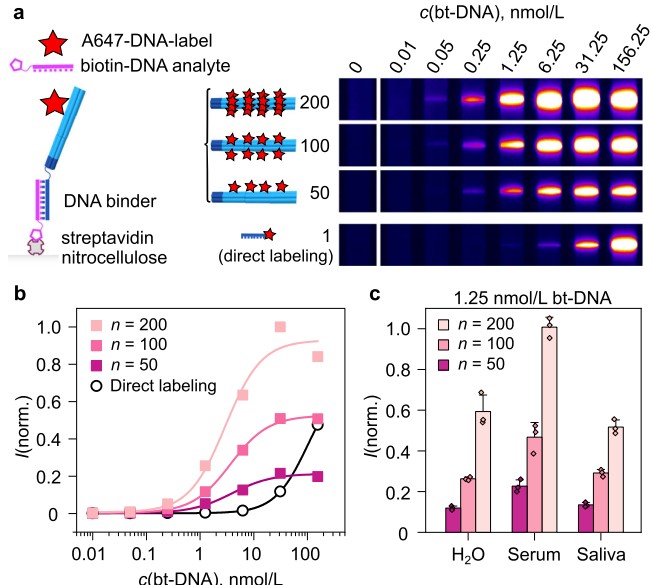

**Fig. 3 | Proof-of-principle detection of biotinylated DNA and adjustable fluorescence signal in lateral flow assays. a** Detection of a biotin-DNA analyte (26-mer DNA oligonucleotide with a 3' biotin) in water. Lateral flow assays were performed using 6HBs mixed with A647-DNA labels at 50-, 100-, or 200-fold excess ($n$) per 6HB. The non-amplified ($n = 1$) reference (bottom panel) contains no 6HB, but instead, A647-DNA that is complementary to the biotin-DNA analyte. The images show A647 fluorescence at the test line after 15 min assay time, captured with a 16 ms exposure time. **b** Normalized fluorescence intensities of the test line images from panel (**a**). Each set of intensity data has been fitted with a four-parameter logistic function. **c** Normalized test line intensities for 1.25 nmol/L biotin-DNA in water, human serum, and saliva, detected with 6HBs with varying A647-DNA amounts. The bar chart displays the mean values ± SD ($n = 3$), with individual data points overlaid. Source data are provided as a Source Data file.

molecules may quench each other when loaded in high density on the 6HBs, the difference between the observed 125-fold amplification factor and the largest possible amplification factor of 180 (equal to the number of handles on the 6HB) is in the expected range. Figure 3b additionally shows that the average fluorescence signal from the 6HBs decreases by 41% when $n$ is decreased from 200 to 100 and by a further 66% when going from $n = 100$ to 50, thus closely following the excess of labels used in the self-assembly reaction.

While the higher label-analyte stoichiometry is the main determining factor for the increased signal and sensitivity at low and mid-range biotin-DNA concentrations, the effects of other experimental factors increase at the highest concentrations. The saturation intensities for each $n$, for instance, depend on the choice of experimental parameters, such as the amount of 6HBs used per strip, the number of analyte-binding sites per 6HB, the applied running protocol, and the capacity of the test line. By adjusting the aforementioned parameters, the response curves of the 6HB signal amplification system can be tuned to fit the most important concentration range of the selected analyte.

Finally, we verified that the signal amplification shown in Fig. 3a–b for biotin-DNA can be reproduced in complex sample matrices. To show this, we compared the detection of biotin-DNA in water, human serum, and human saliva. The lateral flow assays were again run by mixing the sample with the A647-labeled 6HBs before application on the strip, which, in this case, brings the 6HBs into direct contact with each sample matrix before dilution by the running buffer takes place on the test strip. As seen in Fig. 3c, the 6HBs were compatible with all tested sample matrices, and the signal amplification factor could be controlled with $n$, with a slight variation of the fluorescence intensities depending on the type of sample. No test line signal was observed for

blank samples without biotin-DNA, confirming the specificity of the assay in all of the studied sample matrices (Fig. S7).

### Detection of cardiac troponin I

We then constructed a full antibody sandwich-based LFIA for the detection of cTnI. To achieve this, capture antibodies and detection antibodies need to be introduced into the assay. Here, we used a combination of biotinylated capture antibodies ($Ab_{capture}$-biotin) and detection antibody-6HB conjugates ($Ab_1$-6HB or $Ab_2$-6HB) on the same lateral flow test strips with a streptavidin test line as applied for validating the method with biotin-DNA. In addition to detecting cTnI in the A647 fluorescence assay, we combined the 6HB signal amplification with gold-DNA labels for higher sensitivity and an instrument-free readout.

To study the sensitivity and specificity of the cTnI detection, we used serum samples with known cTnI concentrations. For this, we spiked human serum with recombinant cardiac troponin IC complex (cTnIC). cTnIC is one of the three forms of cTnI present in the circulation after an acute myocardial infarction—free cTnI, the binary cTnIC complex, and the ternary complex of troponin I, C, and T[38]. The higher stability of cTnI in the IC complex than in the free form makes cTnIC a reliable concentration standard in assay development. Despite the presence of troponin C, all serum troponin concentrations are reported as the concentration of cTnI. While molar concentrations make it easier to compare the sensitivity of detection between different analytes, cTnI levels are conventionally reported in the literature as mass concentration (typically, as ng/mL or ng/L). We thus use both units of concentration in parallel, and the conversion between molar and mass concentration has been calculated using a molar mass of 23,900 g/mol for cTnI.

The principle of the cTnI detection and the A647 fluorescence assay is described schematically in the left panel of Fig. 4a. To run the assays, a small volume (5 µL) of human serum was first mixed with both the $Ab_1$-6HB conjugate and the $Ab_{capture}$-biotin. The mixture was applied on the sample application pad of the test strip and flushed over the nitrocellulose membrane with running buffer. As an unamplified reference ($n = 1$), the DNA handles on the $Ab_1$-DNA were labeled directly with a complementary A647-DNA oligonucleotide. The smallest detectable concentration with direct labeling was 1.25 nmol/L (Fig. 4a). Test lines at cTnI concentrations below this could not be distinguished from the background fluorescence even when imaging the strips with significantly longer exposure times (from 40 ms up to 1 s). When using $Ab_1$-6HB with $n = 100$, fluorescence at the test line could be detected down to 10 pmol/L concentration of cTnI. This corresponds to *ca.* 100-fold improvement of the LoD. Furthermore, the dynamic range of the assay increased by *ca.* 3 orders of magnitude (0.01–31.25 nmol/L). At the highest tested cTnI concentration, 156.25 nmol/L, the high-dose hook effect, originating from the saturation of capture and detection antibodies by excess analyte, causes a decrease of the test line intensity for both amplified and unamplified strips. Full LFIA strip images are presented in Fig. S8, showing that comparable signal amplification takes place at the control line. In Supplementary Note 8 and Fig. S9, we additionally show that we achieved a comparable detection sensitivity (10 pmol/L) for NfL in serum by changing the detection molecules in the A647 assay to monoclonal anti-NfL antibodies. Notably, this sensitivity is 10-fold higher than what has been previously demonstrated for NfL detection in nanoshell-assisted LFIAs[39].

To run the 6HB assays with gold-DNA labels, we used test strips with dried gold-DNA labels (Fig. 2d) and a running protocol described in Fig. 2e, with the exception that the serum samples were mixed with both the Ab-6HB and the $Ab_{capture}$-biotin before the assay. While the same $Ab_{capture}$-biotin was used on all test strips, here we compared the performance of two Ab-6HB conjugates: $Ab_1$-6HB and $Ab_2$-6HB. A broader screening of different antibody pairings in the A647 assay is

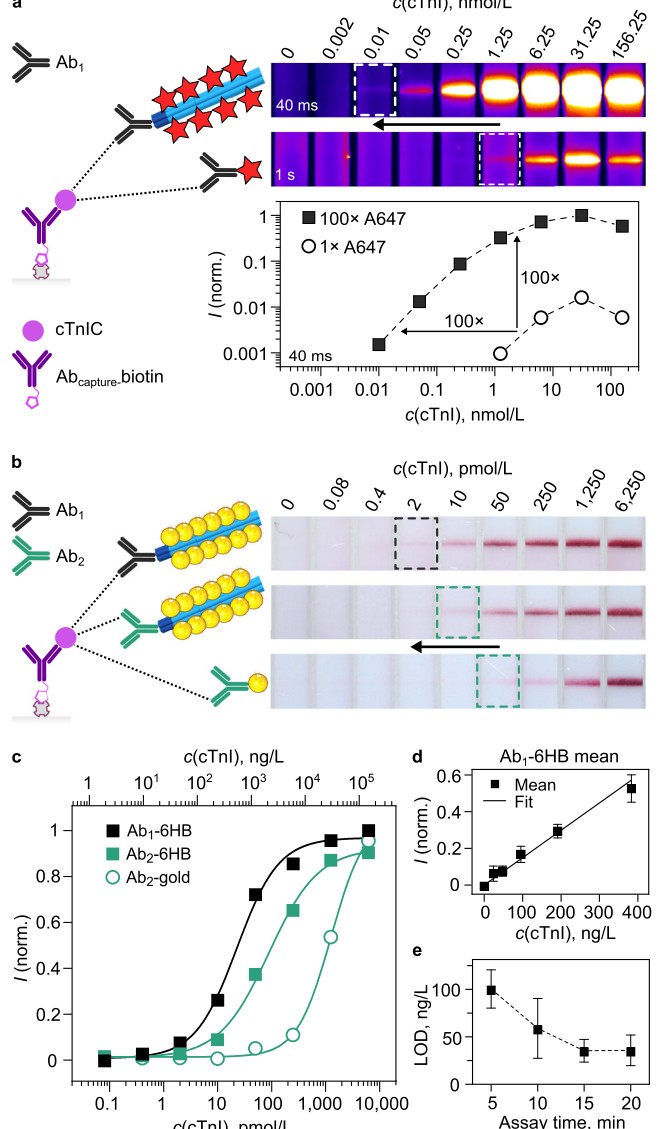

**Fig. 4 | Detection of cTnI in human serum spiked with cTnIC. a** A647 fluorescence assay. The left panel illustrates the 6HB signal amplification complex formed at the test line. The right panel shows test line fluorescence at 15 min assay time for Ab$_1$-6HB with 100 A647-DNA labels and A647-labeled Ab$_1$-DNA. Visual LoDs are highlighted. In the bottom-right panel, test line intensities are plotted against the cTnI concentration. Exposure times for fluorescence image acquisition are shown in the bottom-left corner of each panel. **b** Gold-DNA assay with Ab$_1$-6HB, Ab$_2$-6HB, and Ab$_2$-gold. Test line images at 15 min are shown in the right panel, with visual LoDs highlighted. **c** Binding curves from panel (**b**), fitted with a 4-parameter logistic function. **d** Test line intensities near the LoD for Ab$_1$-6HB. **e** LoD dependence on the assay time for Ab$_1$-6HB. Panels d and e are displayed as mean ± SD (*n* = 3). Source data are provided as a Source Data file.

shown in Fig. S10. As a reference for state-of-the-art sensitivity, we prepared Ab$_2$-gold labels using a passive adsorption protocol. For LFIAs, a 5-μL serum sample was then mixed with the Ab$_2$-gold and the Ab$_{capture}$-biotin before application on the test strip. The amount of gold particles (gold-DNA or Ab$_2$-gold) was identical on all strips.

Figure 4b shows a comparison of the cTnI detection sensitivity of the 6HBs with gold-DNA labels and the Ab$_2$-gold labels. In the presented 1:5 dilution series of cTnIC in serum, the visual LoD of Ab$_2$-gold was at 50 pmol/L (1195 ng/L), concordant with the sensitivity of commercial cTnI LFIAs (Supplementary Note 10 and Fig. S11). By visual LoD, we here refer to the lowest cTnI concentration that produces a visible

test line in the presented dilution series. This LoD could be decreased to 10 pmol/L (239 ng/L), corresponding to a 5-fold sensitivity increase, by conjugating Ab$_2$ to the 6HB (Ab$_2$-6HB). By optimizing the antibody sandwich with Ab$_1$-6HB, the visual LoD could be decreased down to 2 pmol/L (48 ng/L), corresponding to a further 5-fold improvement. Comparison of the inflection points of the sigmoidal fits to the test line intensities in Fig. 4c shows that a 14-fold improvement in assay sensitivity was achieved with the Ab$_2$-6HB, and an overall amplification factor of 55 with Ab$_1$-6HB (inflection points: 23 pmol/L, 89 pmol/L, and 1.26 nmol/L for Ab$_1$-6HB, Ab$_2$-6HB, and Ab$_2$-gold, respectively). In the scope of this study, we did not encounter any issues with specificity related to either nonspecific attachment of our assay components to the test region, or to matrix interference.

Comparison of the signal amplification outcome between the A647-DNA and the gold-DNA labels (Fig. 4a–b) highlights the characteristics of each type of label. In line with the label attachment on the 6HBs shown in Fig. 2c, the signal amplification factor follows the physical constraints of each type of label. The small size of A647-DNA means that the number of labels per origami can be precisely controlled over two orders of magnitude (1–180 labels per 6HB). This was confirmed in the experiments shown in both Figs. 3 and 4a. The larger 40-nm gold-DNA labels can be attached to the 6HBs in lower numbers and likewise, a signal amplification of one order of magnitude was observed (Fig. 4b–c). Even with the lower number of labels per 6HB, the gold-DNA labels provide a high contrast through their high extinction, and in our experiments, they outperformed the A647-DNA labels in terms of sensitivity.

A further study of the LoD of the Ab$_1$-6HB detection was carried out by detecting cTnI with the gold-DNA labels in the low concentration region (1–64 pmol/L, 24–1,530 ng/L, in serum) in three repeated experiments. Figure 4d shows the linearity of the test line signal in this region at 15 min assay time. The data for 5, 10, and 20 min assay times and the details of the linear fits to the data are presented in Figs. S12–S13. As shown in Fig. 4e, the LoD of the assay, defined as LoD = 3.3 × RSD/*b*, where RSD is the residual standard error and *b* is the slope of the linear fit, decreases with assay time. This happens both as more gold-DNA labels bind to the 6HBs and increase the test line signal, and as unbound labels migrate to the liquid absorption (waste) pad and improve the contrast between the test line and the nitrocellulose membrane. Here, a final LoD of 1.5 ± 0.5 pmol/L (35 ± 12 ng/L) was reached in 15 min. It should be noted that the optimal assay time and the LoD are strongly dependent on experimental parameters, such as the test strip materials and size, the arrangement of sample and conjugate pads, the sample volume, and the volume and composition of the running buffer. For instance, following test development procedures of industrial partners, we here used a test strip design for a low sample volume (5 μL). A larger sample volume (e.g., 50–100 μL; a typical range for commercial cTnI LFIAs) could potentially further lower the LoD by increasing the number of analyte molecules per assay (Fig. S11). Put together, our results show that the 6HB signal amplification increases the assay sensitivity and retains the critical short assay time (<20 min) despite the more complex assembly of assay components on the test line.

## Discussion
Due to the exceptional level of control over shape, size, and molecularly precise addressability, DNA origami structures have established their role in academic research as advanced nanoscale measurement tools, programmable devices, and structural scaffolds[22,40,41]. An essential question remaining within the DNA nanotechnology field is, however, how these unique advantages can be leveraged to overcome current technological limitations and provide value in everyday life or commercial applications.

Our signal amplification method presents a general strategy for improving the sensitivity of LFIA tests. One of the key factors leading to

its performance is based on the ease of arranging organic molecules and inorganic particles on DNA origami in precisely controlled amounts. In state-of-the-art LFIAs, the stoichiometry of antibodies to labels is always larger or equal to 1. For 40-nm gold with passively adsorbed antibodies, dozens of antibodies are typically bound to a single label[42]. Moreover, the number of detection antibodies often scales with the total surface area of the labels, i.e., it increases with the amount or size of the labels[6]. In our case, the number of antibodies and label entities per test can be adjusted separately from each other. Here, we have used 1–6 antibodies linked to up to 180 dye molecules or up to 25 40-nm gold labels to boost the amplification factor and, thus, the overall LFIA sensitivity. Generally, adjusting the sensitivity of state-of-the-art assays can be laborious and often requires revisiting the initial antibody selection. Tuning the ratio between labels and detection molecules, in contrast, allows for simple sensitivity adjustment.

A second major benefit of DNA origami arises from its roots in molecular programming. While other methods of nanofabrication always lead to a distribution of sizes and shapes, DNA origami structures are inherently exact macromolecular copies of each other. With its length of 490 nm and width of 8 nm, our DNA origami 6HB can be considered a flexible rod, possibly allowing for reptating, unhindered motion through the porous nitrocellulose matrix[43]. As both the DNA origami and the 40-nm gold particles are small compared to systems that rely on a direct increase in label size, our platform profits from the high diffusivity of its components, which in turn can lead to increased LFIA sensitivity[6].

A more subtle advantage that we played out in our DNA origami design is the placement of antibodies at the ends of the 6HBs, which both gives them high accessibility to the solution and separates them from the labels at the amplification domain. This way, the labels are precluded from sterically interfering with the sandwich formation at the test line. Such design choices ensure that when the number of detection molecules is reduced, all these molecules can remain active. The stoichiometric, distance-, and even orientation-controlled placement of antibodies on DNA origami can additionally enhance target binding. In particular, cooperative binding and avidity effects can be leveraged to improve the sensitivity and specificity of assays[22,23,44,45]. Such strategies hold great potential for advancing the development of future LFIAs.

Next to mere assay performance, it is important to consider the manufacturing costs for DNA origami-based LFIAs. To be widely available, LFIA tests need to be mass-produced at low costs. Consequently, signal amplification technologies aimed at this platform should not increase the costs of materials and production. In our experiments, we applied 12 fmol (77 ng) of DNA origami to each LFIA test. At a cost of 350 EUR per nmol of DNA origami (phosphoramidite solid-phase oligonucleotide synthesis, in-house produced p8634 scaffold), this amounts to an increase of less than one cent of material costs per test. Moreover, in commercial LFIA tests, the detection antibodies are a major source of material costs. As our DNA origami adapter removes the strong link between a number of antibodies and a number of labels, it also allows minimizing the amount of antibodies per test, potentially leading to reduced overall costs of the signal-amplified LFIAs.

In conclusion, we have shown that DNA origami can be integrated into LFIAs for simple, adjustable, and user-friendly signal amplification for the detection of analytes in relevant sample matrices such as water, saliva, and serum. Using cTnI as a benchmark, we performed a direct sandwich-based assay, where the number of labels per cTnI was increased to achieve signal amplification factors of approximately 100 for fluorescent dyes and 14–55 for gold nanoparticles. An LoD of 35 ng/L−or 1.5 pmol/L−was reached in an assay time of 15 min with gold labels, complying with the need for fast, sensitive, and simple PoC test devices. When compared to existing cTnI assays, the obtained sensitivity lies between the low sensitivity of conventional cTnI LFIAs

on the market and the high-sensitivity cTnI laboratory assays with typical LoDs of <10 ng/L[32]. The method is readily adaptable to other biomarkers through antibody selection, as shown in our results for NfL detection (Fig. S9). In addition to the full-length IgG antibodies used here, detection molecules such as aptamers, engineered antibody fragments, and oligonucleotides can be linked to DNA origami structures via established conjugation approaches. Further, our method allows for the simultaneous use of multiple labels in multiplexing assays. Our work presents a step towards the integration of DNA origami nanostructures into commercial, everyday products. Due to the versatility of the approach, it has broad application potential for the development of diagnostic tools and it will enable expanding the range of analytes detectable by conventional LFIA testing.

## Methods

### Preparation of the 6HB signal amplification structure

The 6HB structure and the sequences of the 223 staple oligonucleotides (Fig. S14 and Table S1) were designed using caDNAno2[46] based on the p8634 scaffold. The staple oligonucleotides were purchased from Integrated DNA Technologies. The DNA origami self-assembly reaction was carried out by mixing the scaffold at 50 nmol/L concentration with a 10-fold molar excess of each staple strand in a 1× Tris-EDTA (TE) buffer containing 20 mmol/L $MgCl_2$ and 5 mmol/L NaCl, heating the mixture to 65 °C for 5 min, and cooling to 25 °C over 16 h. The 6HBs were purified from excess staple strands using size-exclusion chromatography with a Bio SEC-5 column (Agilent Technologies) and an ÄKTA Explorer chromatography system (GE Healthcare). The protocol for the purification was adapted from Wagenbauer et al.[47]. The HPLC-purified 6HBs were stored in a 1× TE buffer with 5 mmol/L $MgCl_2$ and 200 mmol/L NaCl at 4 °C. In all of the presented experiments, the 6HBs contained 6 antibody-binding 5' overhangs (TTC ACT ACT TAC CAC TCT ACC) positioned in 2 separate binding domains at the ends of the DNA helices, and 180 label-binding 3' DNA overhangs ($A_{19}$).

### Gel electrophoresis

Agarose gel electrophoresis (AGE) was used for characterizing and gel-purifying 6HBs, antibody-6HB conjugates, and label-6HB complexes. Agarose gels (0.5–1%) were prepared in a 1× Tris-acetate-EDTA (TAE) buffer with 11 mmol/L $MgCl_2$ and stained with SYBR Safe DNA stain (Thermo Fisher Scientific). DNA origami samples were mixed with loading dye (1× loading dye containing 1× TE buffer with orange G and 2.5% Ficoll 400), and the gels were run at a constant 70 V voltage in an ice bath for 60–120 min. The gels were imaged with a Chemidoc MP gel imaging system (Bio-Rad Laboratories) controlled by Image Lab software (Bio-Rad Laboratories, version 6.0.0). The SYBR Safe DNA stain was visualized with UV light. The A647 fluorescence from gels containing A647-labeled 6HBs was imaged using red light excitation (625/30 nm excitation and 695/55 nm detection bandpass filter).

### Gel purification

Gel purification was performed to prepare 6HB samples for TEM. 6HB samples were run on a 1% agarose gel stained with SYBR Safe DNA stain and cut out the band of interest with a razor knife. The liquid containing the samples was squeezed out of the gel pieces between two glass slides and collected into a 1000-µL pipette tip.

### Transmission electron microscopy

For TEM analysis, droplets of the samples (5 µL) were deposited on copper grids (Ted Pella). After an incubation time of 3 min, the rim of each grid was held against filter paper to dab off the liquid. Subsequently, the grids were stained with uranyl formate in two steps. For each sample, two droplets (5 µL) of 2% uranyl formate were deposited on parafilm. In the first step, the uranyl formate solution only quickly washes the grid, followed by dabbing off as described above. In the second step, the grid is left to incubate for 15 s before being dabbed

off. Images were taken with a JEOL JEM 1011 electron microscope at 80 kV.

## Antibody modification

The antibodies for cTnI detection were purchased from HyTest. In the presented experiments, monoclonal mouse anti-cTnIC antibody 20C6cc was selected as the capture antibody ($Ab_{capture}$), and monoclonal rabbit anti-cTnI antibody Y302 ($Ab_1$) and monoclonal mouse anti-cTnI antibody 19C7cc ($Ab_2$) were used as detection antibodies. Monoclonal antibodies 560cc and Y309 for further antibody pair screening (Fig. S10) were likewise purchased from Hytest.

Biotinylation of capture antibodies and subsequent purification by desalting was done with an EZLabel antibody-biotin labeling kit (BioVision) using the provided materials and protocol.

To label detection antibodies with DNA handles, they were first conjugated with a DBCO-PEG$_5$-NHS crosslinker (Iris Biotech) and subsequently with 3'-azide-modified DNA oligonucleotides (GGT AGA GTG GTA AGT AGT GAA-C3-azide, Ella Biotech). In the first step, the antibodies were incubated with a 15-fold molar excess of the crosslinker for 2 h at 4 °C and purified using 40K Zeba spin desalting columns (Thermo Fisher Scientific). In the second step, the antibodies were incubated with a 10-fold molar excess of azide-modified DNA at 4 °C for 20 h and purified using 100 kDa Amicon Ultra centrifugal filters (Merck Millipore). The degree of labeling (DoL) was estimated from the UV–Vis absorbance spectrum of the conjugates, measured using a NanoDrop 1000 spectrophotometer, as

$$DoL = \frac{n(DNA)}{n(Ab)} = \frac{\epsilon_{260}(Ab) - A_{260}/A_{280} \times \epsilon_{280}(Ab)}{A_{260}/A_{280} \times \epsilon_{280}(DNA) - \epsilon_{260}(DNA)} \quad (1)$$

The $\epsilon_{260}(Ab)$, $\epsilon_{260}(DNA)$, $\epsilon_{280}(Ab)$, and $\epsilon_{280}(DNA)$ are the molar extinction coefficients of IgG antibodies and the azide-DNA oligonucleotides at 260 nm and 280 nm, respectively, and $A_{260}/A_{280}$ is the ratio of measured absorbance values at 260 and 280 nm. Sodium dodecyl sulfate–polyacrylamide gel electrophoresis (SDS-PAGE) and native PAGE were used to further characterize the DNA-labeled antibodies and confirm the attachment of the DNA handles (Fig. S1). The labeled antibodies were stored in pH 7.2 phosphate-buffered saline (PBS) at 4 °C.

## Preparation of labels

**DNA loading on gold nanoparticles.** 40 nm gold nanoparticles for DNA loading were purchased from GATTAquant. To modify them with DNA, they were brought to 0.03% SDS in a 0.3× TAE Buffer (pH 3) and mixed with thiol-modified oligonucleotides (5'-HS-T$_{19}$-3', Biomers), resulting in a solution of 30.5 µmol/L T$_{19}$-oligonucleotides and OD 13.7 gold nanoparticles. After incubating the mixture for 15 min at pH 3, the concentration of NaCl was increased to 0.98 mol/L by adding two increments of 5 mol/L NaCl. The low-pH salt aging process of the oligonucleotide/gold nanoparticle solution was followed by incubation at RT for 2 h. After incubation, 1 mol/L NaOH was used to raise the pH from 3 to 8. To remove excess oligonucleotides, the gold nanoparticles were pelleted by centrifugation, the supernatant was removed, and the particles were resuspended in 0.05% SDS. This washing process was repeated 5 times, followed by two rounds of washing with milli-Q water. After the final centrifugation, the concentration of the gold-DNA labels was adjusted to OD 150, and they were stored at 4 °C.

**Antibody-gold conjugation.** 40-nm gold nanoparticles for antibody conjugation were purchased from BBI Solutions. The particles were mixed with 19C7cc antibodies in 10 mmol/L borate buffer (pH 8) at final concentrations of 0.02 mg/mL 19C7cc and OD 1 gold. The mixture was incubated for 20 min at 20 °C and 600 rpm on an orbital shaker. 5% BSA in water (Merck Millipore) was then added to give a BSA concentration of 0.33%, and the mixture was incubated for a further

20 min at 20 °C and 600 rpm. To remove excess antibodies, the gold nanoparticles were pelleted by centrifugation, and the supernatant was removed. The particles were then resuspended in 9 mmol/L borate buffer containing 0.45% BSA. This washing procedure was repeated 6 times. After the final round of washing, the concentration of the antibody-gold labels was adjusted to OD 10 in 8 mmol/L borate buffer containing 1% BSA (pH 8), and they were used in liquid form on LFIAs immediately after preparation.

## Attachment of antibodies and labels on 6HBs

**Preparation of antibody-6HB conjugates.** Antibody-6HB conjugates were prepared by mixing DNA-labeled antibodies with HPLC-purified 6HBs and incubating the mixtures at 4 °C for a minimum of 2 h before use. The antibody-6HB conjugates used in the cTnI immunoassays were prepared by using a molar ratio of 6 antibodies ($Ab_1$-DNA or $Ab_2$-DNA) per 6HB in the reaction mixture. The effect of the molar ratio on the Ab-6HB conjugate assembly was studied with AGE (Fig. S2), and its effect on the cTnI LFIA performance was tested with $Ab_1$-6HB conjugates at molar ratios 0–24, confirming the suitability of the molar ratio of 6 (Fig. S15).

**Characterization of the assembly of gold-DNA and A647-DNA labels on 6HBs in solution.** The DNA origami was added to a solution of gold-DNA labels in a ratio of 300 gold nanoparticles per 6HB. The mixture was incubated for 2 h at RT on a shaker. AGE was used to separate excess gold nanoparticles from the assembled structures, which were then gel purified from a slower migrating band for TEM analysis.

The A647-DNA labels (5'-Alexa Fluor 647-T$_{19}$-3', Eurofins Genomics) were mixed with the 6HBs and incubated for a minimum of 2 h at 4 °C before use. AGE was used to screen the attachment of the labels at molar ratios of 25–200 labels per 6HB (Fig. S6).

## Lateral flow assays

The lateral flow assays were performed on GenLine HybriDetect test strips (Milenia Biotec). All strips contained a nitrocellulose membrane with a streptavidin test line and an anti-rabbit antibody control line. Human serum from male AB plasma for the lateral flow experiments was purchased from Sigma-Aldrich (catalog number H4522). Human saliva was collected and pooled from healthy volunteers. Assay Defender (Candor Bioscience) was used as a running buffer on all test strips. Test strip images in the text are always presented so that the direction of the lateral flow is from bottom to top.

**Biotin-DNA fluorescence assays.** The biotinylated DNA analyte (GGT AGA GTG GTA AGT AGT GAA-C3-biotin) was purchased from Ella Biotech. A 1:5 series of dilutions (156.25 nmol/L–10 pmol/L) of the biotin-DNA was prepared in milli-Q H$_2$O.

The 6HBs used for biotin-DNA detection were prepared with either 50, 100, or 200 A647-DNA labels per 6HB. Identically to the 6HBs used for preparing antibody-6HB conjugates, the 6HBs contained 6 analyte-binding 5' overhangs (TTC ACT ACT TAC CAC TCT ACC) located in 2 separate binding domains at the ends of the DNA helices. A DNA oligonucleotide with an identical sequence to the analyte-binding overhang of the 6HB, but with a 3' A647 modification (TTC ACT ACT TAC CAC TCT ACC-Alexa Fluor 647, Eurofins Genomics), was used to directly label the biotin-DNA with A647.

For running the lateral flow assays, 5 µL of biotin-DNA was first mixed with 3 µL of either the A647-labeled 6HBs or the direct labeling probe. The final concentration of 6HBs in the mixture was 1.5 nmol/L, and the final concentration of A647-DNA labels was either 75, 150, or 300 nmol/L for a molar ratio of 50, 100, or 200, respectively. The final concentration of the direct labeling probe was 150 nmol/L. The mixture was then directly applied on the sample application pad of the lateral flow test strip without

incubation. 30 µL of running buffer was added to the sample application pad below the sample mixture. 5, 10, and 15 min after application of the running buffer, the fluorescence of A647 on the strips was imaged with a Chemidoc MP gel imaging system using red light excitation (625/30 nm excitation and 695/55 nm detection bandpass filter) and a 16 ms exposure time.

For compatibility testing with different sample matrices, 1.25 nmol/L dilutions of biotin-DNA were prepared in milli-Q water, serum, and saliva. Blank samples without biotin-DNA and the 1.25 nmol/L dilutions of biotin-DNA in all three sample matrices were analyzed using the A647 fluorescence assay and an identical assay protocol to the one described above. The experiment was repeated three times.

The fluorescence images were analyzed with Fiji (ImageJ)[48]. Pixel values along the strips were measured, and test line intensities were defined as the maximum intensity value at the test line (i.e., peak height). All collected intensity vs. concentration data were observed to follow symmetrical, sigmoidal trends, which could be fitted with a 4-parameter logistic function,

$$y = A_2 + \frac{A_1 - A_2}{1 + (x/x_0)^p} \qquad (2)$$

where $A_1$ is the response at infinite concentration, $A_2$ is the response at 0, $x_0$ is the inflection point of the curve, and $p$ is the slope factor. The non-linear curve fitting was performed with OriginPro 2023 version 10.0.0.154.

**cTnI immunoassays.** Recombinant human cardiac troponin IC complex (cTnIC) was purchased from HyTest and stored in aliquots at −70 °C. Dilutions of cTnIC (156.25 nmol/L–80 fmol/L as a 1:5 series) were prepared in human serum.

$Ab_1$-6HB and $Ab_2$-6HB conjugates were prepared as described earlier. Additionally, $Ab_1$-6HB conjugates with 100 A647-DNA labels per 6HB were prepared for fluorescence detection by mixing the 6HBs simultaneously with both a 6-fold excess of $Ab_1$-DNA and a 100-fold excess of A647-DNA, and incubating the mixture for a minimum 2 h before use. For direct labeling of $Ab_1$-DNA with A647, it was mixed prior the assay with a 17-fold excess of the TTC ACT ACT TAC CAC TCT ACC-Alexa Fluor 647 oligo (Eurofins Genomics).

To run the fluorescence cTnI assays, 5 µL of cTnIC in serum was first mixed either with 3 µL of A647-labeled $Ab_1$-6HBs or with 3 µL of A647-labeled $Ab_1$-DNA, and then with 1 µL of biotinylated $Ab_{capture}$. The final concentrations in the 6HB mixture were 1.3 nmol/L of $Ab_1$-6HB (containing 7.8 nmol/L $Ab_1$-DNA), 133 nmol/L of A647-DNA, and 13.3 nmol/L of $Ab_{capture}$-biotin. In the direct labeling sample, the final concentration of A647 was 133 nmol/L, and the concentration of $Ab_1$-DNA was 7.8 nmol/L. The mixtures were then applied immediately on the sample application pad of the lateral flow test strips. 60 µL of running buffer was added to the pad upstream of the sample mixture. The strips were imaged and analyzed after 15 min identically to the biotin-DNA assay strips, with the exception that the exposure time was increased to 40 ms for all data sets. The unamplified test strips were additionally imaged with a 1 s exposure time to confirm that the observed LoDs do not change depending on the applied exposure time.

For running cTnI LFIAs with gold-DNA labels, the labels were first applied and dried on the sample application pad of the test strips. For this, the particles were first brought into 1% BSA at OD 25 and incubated for 2 h. After 2 h, the particles were diluted 1:1 with a solution containing 20% sucrose and 10% trehalose. 3.2 µL of the mixture was applied onto the sample application pad and dried for 2 h at 37 °C. The assays were then run using the same running protocol as in the fluorescence assays. The sample mixture was applied on the sample application downstream of the dried gold-DNA labels, and the running buffer was applied upstream of the labels. 5, 10, 15, and 20 min after application of the running buffer, the strips were imaged with a Canon EOS 750D SLR digital camera. Analysis of the test line intensities and 4-parameter logistic curve fitting was performed in the same way as described for the fluorescence assay strips.

To run assays with $Ab_2$-gold labels, 5 µL of cTnIC in serum was first mixed with 4 µL of $Ab_2$-gold labels at OD 10, and 1 µL of biotinylated $Ab_{capture}$ was added. The final concentration of $Ab_2$-gold in the mixture was thus OD 4, resulting in the same number of labels per test as for the tests run with DNA-gold labels. The sample mixture was added to the sample application pad, 60 µL of running buffer was added to the pad upstream of the sample, and the strips were imaged at 5, 10, 15, and 20 min time points.

In addition to defining the visual LoD as the lowest cTnI concentration where a visible test line could be observed in each experiment, the LoD of the cTnI detection with $Ab_1$-6HBs and the gold-DNA labels was further quantified in a separate experiment. 1:2 dilution series of cTnIC between 1–64 pmol/L was prepared in serum. Test strips for the seven cTnIC dilutions and a blank serum sample were run using gold-DNA labels using the protocol described above. The strips were imaged with a Chemidoc MP gel imaging system using white light illumination at 1 min intervals over 20 min after running buffer addition.

The test line intensity values were determined from the acquired images for 5, 10, 15, and 20 min time points with Fiji (ImageJ). The LoD analysis was performed with OriginPro 2023 version 10.0.0.154. For each time point, test line intensities for cTnIC concentrations of 0–64 pmol/L were plotted against the cTnIC concentration. The test line intensities showed linear dependency on the cTnIC concentration in the range of 1–16 pmol/L, or in some cases, between 1–8 pmol/L. The test line intensity values in the linear range were fitted with a linear regression model. The LoDs for each time point were defined as LoD = 3.3 × RSD/$b$, where RSD is the residual standard error of the fit and $b$ is the slope of the linear regression. The LoD values were determined separately for 5, 10, 15, and 20 min assay times in three repeated experiments, averaged, and reported as the mean ± SD.

### Statistics and reproducibility

All data presented in this study, both in the main text and in the Supplementary Information, have been validated to be reproducible through independent experiments. In instances where reproducibility is not explicitly shown (Figs. 2b–c, 3a–b, and 4a–c), the data represent consistent outcomes of $n \geq 3$ independent experiments. In Figs. 3c and 4d–e, data are presented as the mean ± SD of $n = 3$ independent experiments, as stated also in the corresponding Figure captions.

### Reporting summary

Further information on research design is available in the Nature Portfolio Reporting Summary linked to this article.

## Data availability

The data supporting the findings of this study are available within this article and its Supplementary Information and from the corresponding author(s) upon request. The data generated in this study have been deposited in the Dryad database as a Source Data file under accession code https://doi.org/10.5061/dryad.k6djh9whk (ref. 49).

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

## Acknowledgements
We thank Christoph Pauer, Susanne Kempter, and Thomas Nikolaus for TEM support and laboratory assistance. H.I., J.T., P.C.N., and M.J.U. were supported by the Federal Ministry of Economic Affairs and Climate Action (BMWK) through the EXIST Forschungstransfer (03EFWBY304). S.v.R., P.C.N., and M.J.U. also received funding from the Federal Ministry of Education and Research (BMBF) through the GO-Bio initial (031B0982 and 16LW0130) and from the Bavarian State Ministry of Economic Affairs, Regional Development, and Energy through the Medical Valley Award (M4-2007-0005). L.N. and T.L. acknowledge funding from the ERC consolidator grant "DNA Funs" (Project ID: 818635) and from the Deutsche Forschungsgemeinschaft (DFG; German Research Foundation) through the cluster of excellence e-conversion EXC 2089/1-390776260.

## Author contributions
L.N., P.C.N., T.L., and M.J.U. conceived the use of DNA origami for signal amplification on LFAs. L.N. and P.C.N. performed initial DNA origami design and LFIA experiments. S.v.R. and M.U. selected the biomarker for the study. H.I., T.L., and M.J.U. designed the study. H.I. and J.T. prepared samples and performed experiments. H.I. acquired, curated, and interpreted the data. H.I., T.L., and M.J.U. wrote the manuscript with support from all authors.

## Funding

## Competing interests
L.N., P.C.N., T.L., and M.J.U. declare the following competing financial interest: A patent application (PCT/EP2022/065889; Labeling nanostructure for signal amplification in immunoassays and immunoassays using the labeling nanostructure; Tim Liedl, Maximilian J. Urban, Philipp Nickels, Linh Nguyen) has been submitted. The remaining authors declare no competing interests.
