## [Transparent Peer Review file · Nature Communications]

DNA Origami Signal Amplification in Lateral Flow Immunoassays

Corresponding Author: Dr Heini Ijäs

Version 0:

Reviewer comments:

Reviewer #1

(Remarks to the Author)

In this work, the authors reported a DNA-origami based signal amplification method to improve the sensitivity (i.e., limit of detection) of lateral flow immunoassays, using spiked cardiac troponin I in human serum as the target analyte. The high aspect ratio 6HB origami design was selected to showcase the ability of 55 to 125-fold sensitivity improvement with gold nanoparticle and fluorescent dye labels. However, similar level of sensitivity (~10s pg/mL) of troponin I lateral flow immunoassays have already been reported in the literature (<https://www.sciencedirect.com/science/article/abs/pii/S095656631000031X>, <https://pubs.acs.org/doi/abs/10.1021/acs.analchem.7b05410>, <https://www.sciencedirect.com/science/article/pii/S092849310900006X>, <https://www.nature.com/articles/s41598-021-98199-y>). The reviewer thinks this work may not fit the high caliber of Nature Communications.

Reviewer #2

(Remarks to the Author)

Overall assessment: excellent work, well-presented.

Noteworthy result: the authors use DNA origami nanostructures to improve the sensitivity of a lateral flow immunoassay by as much as 125-fold.

Significance: this work would be of interest to scientists in both DNA nanotech and biosensing, as well as having potential for real-world impact (hence the patent filing noted by the authors).

Originality: the most relevant piece of existing literature is probably the paper cited by these authors as ref 9, in which DNA double crossover tiles were used as part of a lateral flow assay. When it comes to originality, the question must therefore be how the origami structure presented here differs from the double-crossover tile assembly. I think the move to origami has yielded a significant performance advantage and there are a number of non-obvious design features of the origami structure that help deliver this, but I think the authors could clarify this by strengthening the discussion of drawbacks of competing methods (ref 7-9), preferably with reference to the LoDs.

Sufficiency: the results do support the claims.

Methodology: sound but I think some extra details should be added to the methods section (see below).

Flaws/queries/points to address:

-fig 1 is a bit small when pdf is printed

-optional: authors could consider adding to fig 1 a sketch showing conventional approach with multiple antibodies per label for visual comparison

-fig 2a - binding domain at far end is not obvious

-fig 3 and elsewhere - why use four parameter function to fit the data? Is this standard for LFTs? Could you actually fit data with fewer parameters? Can authors say more about how the fit parameters are useful?

-pg7 line 184 - is this the explanation for why n=180 doesn't give x180fold improvement? Could this be clearer?

-most important point: fig 4 caption refers to one image having a longer exposure. Surely the exposure time affects the LoD? Authors should clarify when exposure times were varied and when they were kept the same, to demonstrate where data can be properly compared

-fig 4 caption refers to visual LOD, which implies a qualitative assessment of LoD. However methods section suggests that a rigorous mathematical analysis was done. Can the authors clarify?

- methods - if origami is a new design, cadnano output and staples should be provided in SI

-p12 typo - Thermo Fisher not Fischer

-supp fig 8 - why is control band weaker in left-most bands? Was the label used up in the test band? Which way does the flow go? Can the authors say a few more words about what the control is here?

-supp fig 10 - I think the authors did a linear fit to this data (?). If so, shouldn't the fit line be shown?

Overall I think this is a very strong paper.

Dr Katherine Dunn

Reviewer #3

(Remarks to the Author)

In this manuscript entitled "DNA Origami Signal Amplification in Lateral Flow Immunoassay" by Urban et al., they employed DNA origami as a signal amplifier by carrying reporters such as fluorescent dyes or colloidal gold nanoparticles. One advantage of DNA origami is to precisely control the number of detectors and reporters on a single complex, thus to tune the "amplification factor". Although an improved sensitivity has been achieved comparing to protocols without using DNA origami amplifier, what they demonstrated here only represents an incremental improvement to published work (Decorated DNA-Based Scaffolds as Lateral Flow Biosensors. *Angew Chem Int Ed* 62, e202313243 (2023); Net-Shaped DNA Nanostructure-Based Lateral Flow Assays for Rapid and Sensitive SARS-CoV-2 Detection. *Anal Chem* 96, 3291–3299 (2024); Amplification-Free Nucleic Acid Testing with a Fluorescence One-Step-Branched DNA-Based Lateral Flow Assay (FOB-LFA). *Anal Chem* 95, 13605–13613 (2023)), thus lacks significant technical advancement. In addition, the breadth and depth of the current work were rather limited. Utilities on a variety of targets in clinical samples, especially those having no well-established LIFA tests shall be developed and benchmarked to validate the application capability of this method.

Major comments:

1. The benefit of using DNA origami as signal amplifier is not obvious. DNA assemblies such as DNA tubes (Decorated DNA-Based Scaffolds as Lateral Flow Biosensors. *Angew Chem Int Ed* 62, e202313243 (2023)) have been readily coupled with LIFA for signal amplification. It is true that these assemblies are not precisely controlled in size. But why would one need such a precision in size (as origami offered) on rough LIFA tests that rely on naked eyes for signal estimation? Isn't it like killing a chicken with a cow-slaying knife?

2. Sensitivity and specificity are the two most important properties for a LIFA test. In this regard, DNA origami signal amplifier holds no advantage than like DNA tube signal amplifier. On the contrary, due to its limit in size, it may exhibit lower sensitivity than DNA tube amplifiers of unlimited sizes (which can easily grow into dozens of micrometers). In the meantime, DNA origami amplifier offers no superiority in specificity. If multiple different detection molecules being coupled onto one DNA origami to collectively recognize the target, it may aid in improving the detection specificity, which, however, was not demonstrated.

3. Although the integration of origami can enhance the visible signal and does not require complex preparation, origami is not low in cost and necessitates pre-preparation for separation. While the authors mention, ELISA and ECL require specialized equipment, stringent control over reaction conditions, and operation by trained personnel, they do not touch upon fluorescence. There are various fluorescence methods that can amplify signals, and cost-effective methods to enhance fluorescence and acquire signals are currently available.

4. The use of cTnI as the model target for LIFA development lacks justification. It would be considered more useful to target on unmet clinical needs. As the authors stated, "the current high-sensitivity cTnI assays have reached maturity through five generations of optimization", thus there is no clinical need for developing new cTnI assay. In addition, standard cTnI LIFA tests shall be included for comparison to see if the current methods hold any superiority in either sensitivity, specificity, ease of operation etc.

5. Additional targets shall be tested to validate the generality and applicability of the current method. Whether multiple targets can be detected in one LIFA strip? Whether real clinical samples can be detected?

Minor comments:

1. For gel electrophoresis, a good practice is to run samples on the same gel for comparison, instead of assembling gel pieces together as shown in Fig S1, Fig S2, Fig S6.

2. Better provide the design schemes and staple sequences for the DNA origami designs for others to reproduce.

3. Keep the unit for concentration of analytes consistent across the manuscript. For instance, for cTnI, use either ng/L or pmol/L. Plus, L (litter) should be spelled capital.

Version 1:

Reviewer comments:

Reviewer #2

(Remarks to the Author)

I am satisfied with the revisions undertaken by the authors.

However, it seems a bit of a shame to relegate the new results on NfL to the supplementary material. This sounds like a significant finding and the authors may wish to consider (possibly in discussion with the editor(s)) whether it would be possible to move it into the main text. Similarly, Table 1 in the response to referees might merit placement in the SI for easier access, and the normalized LoD values could be mentioned in the main text. In my opinion these are not essential revisions but may enhance the impact of the work.

As far as I am concerned the manuscript can now be accepted for publication by Nature Communications, with or without the minor changes suggested above.

Dr Katherine Dunn

Reviewer #3

(Remarks to the Author)

The authors have addressed part of the suggestions as raised by the reviewer. It is highly recommended that those unaddressed comments being considered seriously to improve the current work to fulfill the high caliber requirement of Nat Commun.

Major concern

1. It remains unclear how the current test would perform on real clinical samples. LFIA tests have been available for decades, a methodology work on its own, without applications, is not sufficient to justify its publication on high profile journals like Nat Commun.

Minor concerns

1. Results in the SI may be brought into main figures to make them more comprehensive and complete. For instance, the detection of NfL, detection of cTnl by commercial strips.
2. A spread sheet or table of staple strand sequences might be provided for the convenience of later users of this technology.

Response to Referees

Title: *DNA Origami Signal Amplification in Lateral Flow Immunoassays*

We wish to thank the reviewers and the editor for their time and thoughts that they have spent with our manuscript and for the valuable comments that helped us to significantly improve this work.

While the reviewers appreciated the functionality and practicality of our approach, several concerns were raised, most pressing ones regarding the advantages of our signal amplification method in comparison to existing methods, the generality of our method, and the choice of cardiac troponin I (cTnI) as a biomarker.

Based on the comments we received, we have made significant modifications to our manuscript addressing the comments and concerns. Importantly, we present new experimental data as suggested by the reviewers:

1. The performance of our cTnI tests is now compared to commercial LFIA from two different manufacturers.
2. We show that by changing the capture and detection antibodies, we can detect neurofilament light chain (NfL) on the same test strips and using the same assay protocol that we developed for cTnI detection.

Below we reproduce, in blue, all comments from the reviewers, with each comment followed by our response along with a description of any associated changes to the manuscript (*italic*). Those changes are highlighted in the resubmitted text in yellow.

We hope that this resubmitted and thoroughly revised version answers all the comments put forth by the reviewers, and that the manuscript in its presented form is suitable for publication in *Nature Communications*.

Please do not hesitate to get in touch if you have any further queries.

Reviewer #1 (Remarks to the Author):

In this work, the authors reported a DNA-origami based signal amplification method to improve the sensitivity (i.e., limit of detection) of lateral flow immunoassays, using spiked cardiac troponin I in human serum as the target analyte. The high aspect ratio 6HB origami design was selected to showcase the ability of 55 to 125-fold sensitivity improvement with gold nanoparticle and fluorescent dye labels. However, similar level of sensitivity (~10s pg/mL) of troponin I lateral flow immunoassays have already been reported in the literature (<https://www.sciencedirect.com/science/article/abs/pii/S095656631000031X>, <https://pubs.acs.org/doi/abs/10.1021/acs.analchem.7b05410>, <https://www.sciencedirect.com/science/article/pii/S092849310900006X>, <https://www.nature.com/articles/s41598-021-98199-y>). The reviewer thinks this work may not fit the high caliber of Nature Communications.

We thank the referee for the critical assessment of our work and for bringing these publications to focus. Please note that we cited mostly reviews that included such references, but we acknowledge that we should have cited and discussed the most relevant articles in our work in more detail.

However, the main point and novelty in the presented work does not lie exclusively in its improved sensitivity towards cTnI. Instead, we introduced a new tool that acts as an adapter between the labels and the detecting molecules. As an adaptor, it can combine all existing tools while providing full control over amplification factor and dynamic range. It overcomes aggregation hurdles and offers excellent modularity and free choice both in targets and labels and thus also in the readout method.

Comparing different papers and their novelty mainly based on the sensitivity that has been demonstrated for cTnI detection would furthermore overlook the general technological advances. In fact, one reason to showcase our method with cTnI testing is specifically because it is such a well-established analyte, and the large body of existing literature on high-sensitivity testing methods enables us to benchmark our approach in comparison to other available signal amplification strategies and assay formats. In other words, the existence of other sensitive cTnI assays does not by itself reduce the impact of our work.

In the revised manuscript, we were now also able to compare our sensitivity to commercial cTnI tests, and emphasize the generality of our method by adding data on NFL detection.

In the table below, we have summarized the main features of the references mentioned by all referees. Most of the articles focus on LFIA label development. As such, new types of labels and detection methods (readers for magnetic particles, fluorescence, SERS, ...) are introduced, each leading to improved sensitivities related to the read-out apparatus. The work that falls closest to ours is the one by Choi *et al.* (2010), where the authors show that signal amplification can be achieved through aggregation of multiple gold nanoparticle labels per analyte at the test line, a method leading to amplification of unspecific signals if not performed in idealized lab-conditions. In this work, cTnI was detectable at ~10 ng/L, which on first sight appears better than our reported LoD (35 ng/L). Please note that i) the authors did not perform a rigorous analysis of their LoD and ii) they applied 20 times more sample per test than we do (100 μ L vs. 5 μ L), which directly scales with the sensitivity. In Table 1, we thus show also the sensitivity after normalization to molecule number (attomoles) per test, and in this aspect our method outperforms all other published works. In our revised SI, we additionally show that the sample volume is directly proportional to the sensitivity of commercial cTnI tests. Regardless, our method is only in its result related to such an "aggregation of labels" approach. Conceptually it differs fundamentally as the DNA origami acts as a new tool in the toolbox of LFIAs as mentioned above. We wish to emphasize that our method

does not compete with the development of labels and is not made obsolete by high-sensitivity detection methods, but is in fact a separate line of development and innovation for LFIA testing. Conversely, in the field of label development, we would not deny the importance of development of a new type of label/reporter that generates a different type of signal on an LFIA strip because another type of label has been already shown to achieve similar sensitivity.

In summary, combining DNA origami with existing or novel labels can bring the LFIA field as a whole forward towards higher sensitivity while retaining simplicity, speed, and low costs that together generate the high value and use of these test devices.

Actions taken: *The Introduction has been thoroughly revised. We have made a clearer distinction between label-based and signal-amplification-based sensitivity enhancement methods in order to provide a better context to our method. The comparison of detection sensitivities was lacking in the manuscript and has been added in the Introduction section.*

We have added several new references on signal enhancement on LFIAs in the manuscript, including the following references highlighted by referee 1: Choi et al. 2010, Xu et al. 2009, Lou et al. 2018, and Bayoumy et al. 2021. (Additionally, Umrao et al. 2024 and Sun et al. 2023, mentioned by referee 3, were included.)

We have rewritten the last paragraph of the Introduction, which introduces cTnI as a proof-of-concept analyte in our study. The revised paragraph now gives a better overview of the existing cTnI testing methods, both in high-sensitivity laboratory testing and in PoC rapid test devices, and hopefully communicates better our reasoning to use cTnI as a benchmarking analyte.

Data on NfL detection has been added in the SI (Supplementary Note 8 and Fig. S9) and it is referenced in the Discussion section. This addition emphasizes the generality of our method outside cTnI testing.

Table 1. Relevant works showing either enhanced sensitivity of cTnI LFIAs or modular LFIA signal amplification strategies.

Reference	Targets	Reported LoD	Reported sample volume	LoD, normalized to moles	Readout method	Detection principle
Our work	cTnI (NfL in SI)	LoD: 35 ng/L (~1.5 pM)	5 μ L	8 amol	Absorbance / Scattering;eye / camera(Additionally fluorescence, applicable to other labels)	DNA origami adapter with AuNP
[14] Schulte et al. (2023) "Hybridization Chain Reaction..."	SARS-CoV-2; nucleocapsid protein (N) & RNA	200.000 copies/mL, 10 ⁶ copies/mL 100-N-Prot/Virion	300 μ L	10 amol N-Prot	Absorbance; eye/camera	Hairpin Chain Reaction (HCR) and carbon black (CB)
[33] Bayoumy et al. (2021) "Sensitive and quantitative..."	cTnI	LoB: 8.4 ng/L and LoD: 30 ng/L	25 μ L	31 amol	Upconverting fluorescence (Upcon reader device (Labrox Oy, Turku, Finland))	Upconverting (UC) NPs
[31] Xu et al. (2009) "Development of lateral flow..."	cTnI	~ 10 ng/L	100 μ L	42 amol	Magnetic Assay Reader (MAR, Magnabiosciences)	Superparamagnetic nanobeads

[16] Choi et al. (2010) "A dual gold nanoparticle..."	cTnl (myoglobin and IL-5 in SI)	~ 10 ng/L (~10 µg/L and ~100 ng/L)	100 µL	42 amol	Absorbance / Scattering; eye / camera	AuNP aggregation at test line
[32] Lou et al. (2018) "Fluorescent Nanoprobes..."	cTnl	32 ng/L	100 µL	134 amol	Fluorescence ("Fluorescent quantitative immunoassay analyzer")	Oriented Abs and fluorophores on polystyrene spheres
[15] Brannetti et al. (2023) "Decorated DNA-Based Scaffolds ..."	dinitrophenol, Digoxigenin, Thrombin, MUC1, EGFR	> 0.3 nM for all targets in buffer (but 0.05 nM for thrombin in saliva?)	15 µL	4500 amol	Fluorescence (ChemiDoc MP imaging system)	Co-assembly of DNA tubes tiles (not origami)
[12] Umrao et al. (2024) "Net-shaped..."	SARS-CoV-2/ trimeric spike protein (TSP)	1.7 nM TSP (or 10 ³ viral copies/mL)	100 µL	150000 amol TSP	Absorbance / Scattering; eye / camera	Au shell NP coated with DNA net
[11] Sun et al. (2023) "Amplification-Free Nucleic Acid..."	DNA/RNA SARS-CoV-2 pseudovirus	300 copies/mL	Not specified	n.a.	Fluorescence (fluorescence card reader)	Branched DNA and FITC beads (not origami)

Reviewer #2 (Remarks to the Author):

Overall assessment: excellent work, well-presented.

Noteworthy result: the authors use DNA origami nanostructures to improve the sensitivity of a lateral flow immunoassay by as much as 125-fold.

Significance: this work would be of interest to scientists in both DNA nanotech and biosensing, as well as having potential for real-world impact (hence the patent filing noted by the authors).

Originality: the most relevant piece of existing literature is probably the paper cited by these authors as ref 9, in which DNA double crossover tiles were used as part of a lateral flow assay. When it comes to originality, the question must therefore be how the origami structure presented here differs from the double-crossover tile assembly. I think the move to origami has yielded a significant performance advantage and there are a number of non-obvious design features of the origami structure that help deliver this, but I think the authors could clarify this by strengthening the discussion of drawbacks of competing methods (ref 7-9), preferably with reference to the LoDs.

Sufficiency: the results do support the claims.

Methodology: sound but I think some extra details should be added to the methods section (see below).

We thank the referee for the positive evaluation of our work and in particular for pointing out the significant benefits of our approach that are partly relying on non-obvious design features that we missed

to emphasize in the previous version. Based on the comments of all three referees, it has become obvious that the discussion of existing/alternative LFIA methods lacked the required depth in the original manuscript.

Actions taken: *We now discuss the added benefits resulting from all design features of the DNA origami signal amplification structure (cf. response to reviewer #3) in more detail in the Discussion. Next to refs 7–9, we further included a comparison with (most of) the works summarized in the table presented in the responses to reviewer #1.*

Flaws/queries/points to address:

-fig 1 is a bit small when pdf is printed

-optional: authors could consider adding to fig 1 a sketch showing conventional approach with multiple antibodies per label for visual comparison

-fig 2a - binding domain at far end is not obvious

We thank the reviewer for these suggestions. We have increased the size of Fig. 1 and we have revised Fig. 2a to underline the existence of two binding domains. We decided against adding a comparison to conventional approaches in Fig. 1. Given the breadth of published methods that partly differ only in details, we fear that reducing this diversity to one conventional approach could be viewed as an oversimplification. Instead, we drastically expanded the discussion of existing methods in our introduction.

Actions taken: *We have revised Fig. 2a and included “binding domain 1” and “binding domain 2” into the schematic of the 6HB. In the original manuscript, Fig. 1 was scaled down slightly from its original single-column width. The figure is now presented in the original, larger size. We have decided to still exclude the state of the art schematic and focus the message on introducing the concept of DNA origami signal amplification.*

-fig 3 and elsewhere - why use four parameter function to fit the data? Is this standard for LFTs? Could you actually fit data with fewer parameters? Can authors say more about how the fit parameters are useful?

We thank the referee for this question and for bringing to our attention that our fitting methods and the interpretation of the fitted curves were not adequately described.

Functions for common probability distributions are sigmoidal, and they are regularly used to model trends in immunoassays, such as LFTs and ELISA, using non-linear curve fitting (Hill equation or 4- and 5-parameter logistic equations). The equation of a 4-parameter logistic fit is $y = A_2 + (A_1 - A_2) / [1 + (x/x_0)^p]$ and its parameters are response at zero concentration (A_2), response at infinite concentration (A_1), inflection point (midpoint concentration) (x_0), and the slope factor (p). 4 parameters are thus required for describing symmetric sigmoidal data, and the choice to include a fifth parameter can be made if the model needs to additionally describe asymmetry.

In the manuscript, the fits are used mainly for visualizing the trends in the data. However, in Fig. 4c, we have additionally defined the sensitivity increase achieved with the antibody-6HB conjugates as the difference between the inflection points of the fitted curves. Although this is a central aspect of the interpretation of the data, it was accidentally not mentioned in the text.

Actions taken: *We have now described the fit in more detail in the Methods section. In the Results section, we now provide an explanation of how the amplification factors in Fig. 4c were defined as the differences between the inflection points of the fitted curves.*

-pg7 line 184 - is this the explanation for why n=180 doesn't give x180fold improvement? Could this be clearer?

We thank the reviewer for pointing out this unclear explanation.

Action taken: *We have added the following explanation in the Results section: "When taking into account that the number of accessible DNA extensions at a given site of a DNA origami structure ranges from ~ 70 to 90 % [34] and that dye molecules may quench each other when loaded in high density on the 6HBs, the difference between the observed 125-fold amplification factor and the maximum expected amplification factor of 180 (equal to the number of handles on the 6HB) is in the expected range."*

-most important point: fig 4 caption refers to one image having a longer exposure. Surely the exposure time affects the LoD? Authors should clarify when exposure times were varied and when they were kept the same, to demonstrate where data can be properly compared

We thank the referee for this question, and we acknowledge that the description of the effect of the exposure time used for capturing the images has required a better explanation.

In each fluorescence experiment, where the detection sensitivity with different samples is compared to each other, the exposure time of the fluorescence images is kept constant between data sets to obtain comparable intensity vs. concentration values. For the data presented in Fig. 4a, this exposure time is 40 ms. For the strip images with unamplified fluorescence, we show 1 s exposure time in place of 40 ms. Our reasoning was to show additional proof that the fluorescence signal of the test line cannot be distinguished from the background fluorescence below the visual LoD of 1.25 nmol/L, even at prolonged exposure times. In other words, the LoD does not change (to a significant degree) depending on the chosen exposure time, and we want to present this in Fig. 4.

Actions taken: *We have removed the explanation of the exposure times in the caption of the Fig. 4a and instead focused on providing a better explanation of the 40 ms vs. 1 s exposure times in the text (Results section). "The smallest detectable concentration with direct labeling was 1.25 nmol/L (Fig. 4a). Test lines at cTnI concentrations below this could not be distinguished from the background fluorescence even when imaging the strips with significantly longer exposure times (from 40 ms to up to 1 s)."*

-fig 4 caption refers to visual LOD, which implies a qualitative assessment of LoD. However methods section suggests that a rigorous mathematical analysis was done. Can the authors clarify?

We thank the referee for pointing out this ambiguity in the description of the LoD analysis.

The aspect that was not explained properly is that we refer to two different types of LoDs: First, for most experiments, we refer to the “visual LoD”, which in this case refers to the lowest analyte concentration that produces a visible test line in the experiment in question. For instance, this is 2 pmol/L for the cTnI test strips and Ab1-6HB shown in Fig. 4b, but it is obvious that the actual LoD of the assay lies somewhere between the data points 0.4–2 pmol/L. For cTnI detection, we have thus additionally defined the LoD in more detail with a separate experiment and mathematical analysis and ended up with the value of 1.5 pmol/L.

Actions taken: *We added the definition of visual LoD as “the lowest cTnI concentration where a visible test line can be observed in the presented dilution series” in both the results and the methods section. The wording regarding the visual LoDs and the mathematically determined LoDs has been generally improved in the Results section to make the difference clearer to the reader.*

- methods - if origami is a new design, cadnano output and staples should be provided in SI

Absolutely. We now present the cadnano design output, including the DNA sequences, in the SI (Fig. S14).

-p12 typo - Thermo Fisher not Fischer

Thank you, corrected actually in two instances.

-supp fig 8 - why is control band weaker in left-most bands? Was the label used up in the test band? Which way does the flow go? Can the authors say a few more words about what the control is here?

We thank the referee for bringing up these questions. We always present test strip images so that the direction of the lateral flow is from bottom to top. This information was indeed missing in the manuscript. Importantly, also the control line intensities can inform us about crucial aspects of our binding behavior. To our understanding, two phenomena are contributing to the changes in the control line intensity at higher cTnI concentrations:

- As suggested by the referee, at high cTnI concentrations (particularly visible at 31.25 nmol/L cTnI), most of the DNA origami added on the strip (4 nmol/L) will be bound to cTnI and captured at the test line, thus reducing the amount of origami at the control line.

- The so-called hook effect emerges in our experiment at 156.25 nmol/l cTnI; in this case, the concentration of cTnI is far greater than that of the 6HBs and the antibodies. Excess cTnI molecules are able to separately saturate both the detection antibodies on the 6HB and the capture antibodies on the test line, thus preventing sandwich formation at the test line. As a result, the intensity at the test line decreases, leading to a hook-type shape of the curve. Consequently, more 6HBs are carried over the test line and captured at the control line, whose intensity is recovered.

In principle, the onset of these effects could be tuned by adjusting the amount of DNA origami and the stoichiometry of antibodies and labels.

Actions taken: We added an explanation of the two types of effects in Fig. S8 caption. In addition, as the reduction of test line intensity caused by the second effect (the classic high-dose hook effect) is also visible in the fluorescence test strip images in Fig. 4a, we have added a short description of the hook effect and the dynamic range of the strips in the results section.

We have included a notion both in the methods section of the main text and in all relevant SI figures that strip images are presented with the direction of the lateral flow from bottom to top. The capture molecules at the control line (anti-rabbit antibody) and test line (streptavidin) have been added into the figure caption of Fig. S8 and S9.

-supp fig 10 - I think the authors did a linear fit to this data (?). If so, shouldn't the fit line be shown?

Overall I think this is a very strong paper.

Dr Katherine Dunn

We thank the referee for this suggestion for improving the presentation of our data.

Actions taken: Following the suggestion, we added a new SI figure (Fig. S13) that shows both the intensity vs. concentration data and the linear fits made to the data. Because the fitting for determining the LoD values for each time point (5, 10, 15, 20 min) was done separately for each repeated experiment (3 repeats), all 12 data sets with the fits are shown separately. The Fig. S10 (now Fig. S12), where intensity values have been averaged from the three repeated experiments, has not been modified.

Reviewer #3 (Remarks to the Author):

In this manuscript entitled “DNA Origami Signal Amplification in Lateral Flow Immunoassay” by Urban et al., they employed DNA origami as a signal amplifier by carrying reporters such as fluorescent dyes or colloidal gold nanoparticles. One advantage of DNA origami is to precisely control the number of detectors and reporters on a single complex, thus to tune the “amplification factor”. Although an improved sensitivity has been achieved comparing to protocols without using DNA origami amplifier, what they demonstrated here only represents an incremental improvement to published work (Decorated DNA-Based Scaffolds as Lateral Flow Biosensors. *Angew Chem Int Ed* 62, e202313243 (2023); Net-Shaped DNA Nanostructure-Based Lateral Flow Assays for Rapid and Sensitive SARS-CoV-2 Detection. *Anal Chem* 96, 3291–3299 (2024); Amplification-Free Nucleic Acid Testing with a Fluorescence One-Step-Branched DNA-Based Lateral Flow Assay (FOB-LFA). *Anal Chem* 95, 13605–13613 (2023)), thus lacks significant technical advancement. In addition, the breadth and depth of the current work were rather limited. Utilities on a variety of targets in clinical samples, especially those having no well-established LIFA tests shall be developed and benchmarked to validate the application capability of this method.

We appreciate the reviewer’s comments acknowledging the precise control over amplification and the improved sensitivity that the DNA origami approach offers. At the same time, this reviewer raises concerns about the significance of our technical advancements (further discussed under major comment 1) and suggests measures to widen the “breadth and depth” of our work.

We took this criticism to heart and performed new experiments demonstrating the modularity and generality of our method. We report in our revised manuscript results on neurofilament light chain (NF-L / NfL), a marker for neurodegenerative diseases and ischemic stroke. For NfL diagnostics, currently only ELISA, SiMoA and comparable advanced tests are available. Without optimization of our protocols and using commercial NfL antibodies, our DNA origami test reaches a sensitivity of 10 pM in serum, thus outperforming NfL detection with nanoshell-assisted LFIA 10-fold (Zhang et al. 2013, DOI: 10.1109/EMBC40787.2023.10340109) (to our knowledge the only NfL - LFIA results). Our current test is already sensitive enough for use in rapid diagnostics of liquorrhea, in cases of skull brain trauma that include skull base fractures. This indication requires a fast therapy decision based on the detection of cerebrospinal fluid in the nasal swabs. For testing of cerebrospinal fluid (CSF), we are able to cover the full spectrum of levels for healthy (5–30 pM) and diseased individuals (> 30 pM; note that these are rough values that are also strongly age dependent) in dementia and stroke. In the long term, we aim to improve sensitivity further and address clinical unmet needs in the field of rapid blood tests for neurodegenerative diseases and stroke. While our current sensitivity is still one order of magnitude too high for detecting elevated levels in dementia patients and in all stroke patients, it reaches the NfL levels in blood plasma of patients with severe neuro-axonal injuries (Sanchez et al. 2022, DOI: 10.3389/fneur.2022.841898; Ahn et al. 2022, DOI: 10.1097/MD.00000000000030849).

We also included the references Umrao *et al.* (2024) and Sun *et al.* (2023) to the introduction section of the text.

Below, we discuss the reviewer’s specific comments.

Major comments:

1. The benefit of using DNA origami as signal amplifier is not obvious. DNA assemblies such as DNA tubes (Decorated DNA-Based Scaffolds as Lateral Flow Biosensors. *Angew Chem Int Ed* 62, e202313243 (2023)) have been readily coupled with LIFA for signal amplification. It is true that these assemblies are not precisely controlled in size. But why would one need such a precision in size (as origami offered) on rough LIFA tests that rely on naked eyes for signal estimation? Isn't it like killing a chicken with a cow-slaying knife?

We thank the referee for raising these critical considerations that helped us to strengthen the advantages of using DNA origami on LFIA. In the Discussion section of the manuscript, we now write in greater detail about the specific benefits of applying the DNA origami method on the LFIA format in place of alternative, often “simpler” methods. We share the viewpoint of the referee that this aspect should always be critically evaluated when presenting novel applications of DNA origami.

The referee has also highlighted the work by Brannetti *et al.* (2023) as the study with the highest overlap with our work. We will also address the technical differences between this specific work and our work in our response to comment 2 below and we summarize the main practical differences here: the sensitivity in the work of Brannetti *et al.* remained orders of magnitude worse than in ours (> 0.3 nM vs. 1.5 pM); their test requires extended pre-mixing of components and longer running times (5 min + 30 min vs. 15 min); their read-out is by fluorescence microscope or reader, a detection method generally providing higher sensitivity, while we achieve 1.5 pM with the naked eye; and related to this, they do not show nor discuss whether their method is applicable to nanoparticle labels, such as colloidal gold. The use of standard gold labels requires assembly of the signal amplification complex on the strip and thus introduces an additional molecular assembly process into the signal amplification. Finally, our method offers easy and precise control over the amplification factor, which is of interest to medical partners, who require sensitivity in a certain range that often is not at the highest sensitivity level. All these successful development steps in our work cannot be called merely incremental.

The main question here is about the specific advantages of the controlled size, which we acknowledge was perhaps not backed up with solid enough argumentation. Moreover, our focus on this one advantage of our device was unnecessarily narrow, leading us to lose sight of additional advantages arising from the exquisite design abilities of DNA origami. As also suggested by this referee in comment no. 2 below, DNA origami offers unique possibilities in molecular precision and engineering, such as combining multiple detection molecules to a single DNA origami. We have not yet explored these exciting possibilities of the DNA origami method that go beyond the scope of this work.

Generally, uncontrolled growth or polymerization of components can lead to aggregation in LFA assays. Hence we are convinced of the benefits of our large – but controlled in size – adapter devices that offer space for a defined number of antibodies and labels. Next to reduced aggregation and improved mobility on the membranes, the controlled size implicitly also allows us to control the amplification factor. We expand on these arguments in the reply to comment 2 below.

The controlled size and shape of DNA origami is also linked to the precisely controlled number and spatial position of antibodies and binding sites for labels. We found, for example, that the location of the antibodies and the labels on the origami structures are crucial for their performance. For example, we ensure that the antibodies are readily accessible for antigen detection and are not e.g. sterically blocked by the gold nanoparticle labels, which is why the antibodies are at the ends of the 6HBs and there is a separation between the antibodies and the signal amplification domain. We agree that such design features are not instantly obvious but add up to higher sensitivity, which we now explain in the Discussion

section. In contrast, in the tile-based assembly used by Brannetti *et al.*, the number and location of the detection and reporter elements can only be controlled through their relative numbers in the reaction/polymerization mixture. We speculate that this is one of the reasons for the far worse sensitivity of the Brannetti *et al.* devices.

Actions taken: *This comment is addressed in both the revised Introduction and Discussion sections. Alternative LFIA signal amplification strategies are now presented in a more comprehensive manner in the Introduction, including several new references. In the Discussion section, the specific advantages of DNA origami structures (precisely controlled size, control over stoichiometry, and control over location of detection molecules and labels) in the signal amplification are discussed more broadly.*

2. Sensitivity and specificity are the two most important properties for a LIFA test. In this regard, DNA origami signal amplifier holds no advantage than like DNA tube signal amplifier. On the contrary, due to its limit in size, it may exhibit lower sensitivity than DNA tube amplifiers of unlimited sizes (which can easily grow into dozens of micrometers). In the meantime, DNA origami amplifier offers no superiority in specificity. If multiple different detection molecules being coupled onto one DNA origami to collectively recognize the target, it may aid in improving the detection specificity, which, however, was not demonstrated.

The referee raises here two concerns: first, that our adapters that are limited in size may exhibit lower sensitivity as compared to the DNA tube LFIA by Brannetti *et al.* and second, that assay specificity, which is crucial in all immunoassay development for avoiding false positive signals, has not been addressed in the work.

Our response to comment No. 1 above addresses the practical differences of our method and the DNA tube method. One detail of their work is that technically no immunoassays with antibodies are demonstrated, as all sandwich immunoassays in the paper are with the antibody in the middle, not on the strips nor the labels.

In addition, we have to argue that the comparison of our data to the published data by Brannetti *et al.* does not support the claim that the DNA tubes would exhibit higher sensitivity because of their unlimited sizes. As also summarized in Table 1, the LoD values presented by Brannetti *et al.* for the fluorescent DNA tubes are > 0.3 nM for all tested targets in buffer (including thrombin; interestingly a higher sensitivity is reached for thrombin in saliva: 0.05 nM), while we reach ~ 10 – 50 pM LoDs in serum with fluorescence detection and 1.5 pM with the naked eye, thus orders of magnitude better. Brannetti *et al.* report an average length of DNA tubes at ca. 7 μm , showing in the SI the considerable distribution of lengths from 0.5 μm to >10 μm . Thus, even though the DNA tubes are on average over 10 times longer than the ~ 0.5 μm long 6HBs in our work, the sensitivity is far worse. This indicates that the size of the applied tube by itself is not the defining factor in sensitivity. Regardless of these points, in case the size of the signal amplification structure were to be limit the sensitivity of the assays, DNA origami could easily be produced in virtually any sizes both through the use of orthogonal scaffolds (Engelhardt *et al.* 2019, DOI: 10.1021/acsnano.9b01025) or multimerization of origami structures.

We thank the referee for pointing out that the discussion about assay specificity was missing in the manuscript. We are glad to report that we have not run into specificity issues with our spiked serum samples (we obtain clear zero samples). Additionally we performed a mini study with spiked blood samples of healthy patients, also there the zero samples are clear (Figure 1).

Figure 1. Plasma from blood samples of four healthy individuals was separated and spiked with cTnl.

We agree that in addition to the presented approach of simple sandwich-based immunoassays with one detection and one capture antibody, the DNA origami platform would allow for much more complexity in the arrangement of detection molecules and labels. We believe that in addition to controllable signal amplification, this can provide sophisticated methods for tackling both sensitivity and specificity problems in immunoassays (and thus avoiding the amplification of nonspecific signals and noise). This could be carried out for instance through the approach suggested by the referee, *i.e.* by combination of multiple detection molecules on a single DNA origami structure, potentially at varying distances and densities. This will be a study on its own and falls outside the scope of this manuscript.

Actions taken: *As noted above, we have expanded the manuscript with a more detailed explanation of the differences between previously published methods and our work. We have also added a statement about the specificity of our assay in the Results section and the outlook to tackle specificity issues with further improvement of e.g. antibody placement in the Discussion section.*

3. Although the integration of origami can enhance the visible signal and does not require complex preparation, origami is not low in cost and necessitates pre-preparation for separation. While the authors mention, ELISA and ECL require specialized equipment, stringent control over reaction conditions, and operation by trained personnel, they do not touch upon fluorescence. There are various fluorescence methods that can amplify signals, and cost-effective methods to enhance fluorescence and acquire signals are currently available.

We thank the reviewer for bringing up these points. We like to note that against intuition, DNA origami could even decrease the cost of LFIA tests. We have calculated in our manuscript the material costs of DNA origami per test strip and obtained a value of one cent of added material costs per test (highlighted now in yellow). Obviously, this does not include manufacturing and development costs, but taking economy of scales into account, this would only amount to additional cents. On the other hand, DNA origami allows us to minimize the amount (and cost) of antibody per test while still using the cheap and well-established colloidal gold labels. Regardless, any sensitivity enhancement method for LFIAs is likely to come with some increased costs, particularly for new labels that are both more expensive to manufacture plus they need a reader. Again we note that reader-free detection can be essential in many settings.

As we have shown a considerable amount of fluorescence data in the manuscript, we believe that the comment about not touching upon fluorescence mainly refers to a lack of general discussion about fluorescence-based LFIAs in the text (introduction and discussion). We thank the referee for bringing to our attention that there was an imbalance between the amount of presented fluorescence data and the amount of discussion about the technique. In short, a major feature of the DNA origami signal amplification system is that it is applicable to all labels, including fluorescent molecules, but just as well to fluorescent or magnetic beads.

Actions taken: *In our revised Introduction, we now distinguish sensitivity enhancement methods based on label development and signal amplification more clearly from each other. Fluorescent labels are acknowledged as an effective method for developing higher sensitivity LFIAs through brighter labels.*

4. The use of cTnI as the model target for LIFA development lacks justification. It would be considered more useful to target on unmet clinical needs. As the authors stated, “the current high-sensitivity cTnI assays have reached maturity through five generations of optimization”, thus there is no clinical need for developing new cTnI assay. In addition, standard cTnI LIFA tests shall be included for comparison to see if the current methods hold any superiority in either sensitivity, specificity, ease of operation etc.

We thank the referee for this feedback, which shows that in our original manuscript, we did not succeed in introducing the choice of cTnI in a convincing and coherent manner. As the referee has realized, the main point in the presented work is not improved sensitivity towards cTnI. Instead, we introduced a tool that acts as an adapter between the labels and the detecting molecules and can therefore easily be applied with varying targets. The main reason to showcase our method with cTnI testing is specifically because it is such a well-established analyte, and the large body of existing literature on high-sensitivity testing methods enables us to benchmark our approach in comparison to other available signal amplification strategies and assay formats.

We thus hope that the revised introduction of the manuscript communicates the reasoning behind the choice of the biomarker clearly to the reader. Furthermore the clinical need for more sensitive tests is still true for the PoC, even more so in settings without fast access to laboratories, but in the original wording of the Introduction, laboratory tests and PoC tests were not differentiated well enough from each other.

In the second part of the comment, the referee states that the comparison of the advantages (or superiority) of our method in comparison to existing tests has been insufficient, to which we have reacted by performing new experiments.

In the original manuscript (Fig. 4b–c), we have already shown a comparison of DNA origami signal amplification and conventional antibody-gold conjugates in cTnI detection. This comparison is shown to demonstrate that we are superior in sensitivity. Nevertheless, we agree with the referee that our improved sensitivity is demonstrated in a more convincing manner by presenting a comparison to existing, commercial assays. We have now included a comparison of our method to standard cTnI LIFA tests from two different manufacturers (Fig. S11). The comparison shows that the commercial assays display very similar sensitivity to our in-house prepared state-of-the-art antibody-gold conjugates, and that the reported 55-fold amplification factor (or sensitivity enhancement) with DNA origami also applies to the commercially available assays.

Additionally, there is now also more discussion in the introduction about alternative sensitivity enhancement and signal amplification strategies, which should give more context to the sensitivity enhancement achieved by our method.

Regarding the “unmet clinical need”, our answer to the next comment is also relevant.

Actions taken: *Comparison of our method to commercial cTnI LFIA tests from two different manufacturers has been added into the SI (Supplementary Note 10 and Fig. S11) and referenced in the Results section. The Introduction has been thoroughly revised and now includes a clearer reasoning for the use of cTnI in the experiments. More information on the existing methods and their sensitivities has been added in the revised Introduction, which should aid the reader in comparing our work to other methods and products.*

5. Additional targets shall be tested to validate the generality and applicability of the current method. Whether multiple targets can be detected in one LIFA strip? Whether real clinical samples can be detected?

We thank the referee for these questions and agree that they are all highly relevant in terms of further validation of our method and test development. We also acknowledge that the claims of the generality and applicability of our method were not sufficiently backed up by data in the original manuscript.

We are thus confident that the addition of the data on NfL detection (Supplementary Note 8 and Fig. S9), which we have described also below the first comments of the referee, sufficiently strengthens this argumentation in our work.

Already in the original version of the manuscript, we have shown compatibility with different sample matrices (buffer, serum, saliva), analyte binding strategies (DNA hybridization and antibody-based sandwich immunoassays), and labels (colloidal gold and fluorescence combined with a demonstration of adjustable signal intensities). After including NfL as a second analyte, we believe that these experimental data together demonstrate that our system is as modular and applicable as we claim.

We fully agree that the introduction of clinical samples would considerably strengthen our work. But, while interesting also to us in follow-up work, we would like to argue that both the topics of multiplexing and the measurement of clinical samples fall outside of the scope of this work. In addition to significant workload, one reason for this is that from a methodological point of view, advancing from the current state of the work to detecting multiple targets or measuring clinical samples is mainly a question of test strip engineering. The most critical aspect of multiplex test development would be in the engineering of the nitrocellulose membrane of the test strip in a way that ensures that multiple test (detection) lines or zones are optimally designed and located to not e.g. compete on the available labels. Measuring clinical samples would likewise be expected to provide meaningful data only after implementation of a test strip design that can take up larger sample volumes and where all assay components are ideally dried on separate conjugate pads beforehand. In this study, we chose to demonstrate our method on commercial strips purchased from Milenia Biotec, which on one hand offer a versatile test platform for different analytes and labels, but on the other hand heavily limit e.g. the sample type and volume, and do not enable multiplexing.

For now, we hope that we have sufficiently justified the current scope of the work as a study that focuses on introducing the signal amplification technology and providing sufficient proof (through presenting

compatibility with different analytes, sample matrices, and labels) that the technology is also further generalizable to such future applications.

Actions taken: *We have added a new data set on NfL detection (Supplementary Note 8, Fig. S9, and Supplementary Methods). These results are referenced in the Results section of the main text to justify our claim that our method can be easily adapted to different analytes.*

Minor comments:

1. For gel electrophoresis, a good practice is to run samples on the same gel for comparison, instead of assembling gel pieces together as shown in Fig S1, Fig S2, Fig S6.

We thank the referee for raising this point. Figures S1a, S1b, S2, and S6 each in fact present a single gel, of which only a part of the gel lanes are shown, although this was not indicated in the figure captions. In the case of Figures S1a and S6, irrelevant gel lanes were cropped out of the final figure. In Figures S1b and S2, separating the gel image into smaller fragments was done for aesthetic reasons to better distinguish different samples from each other.

Actions taken: *We removed the cropping from Figures S1b and S2 to avoid giving a false impression about the processing of the gel images. Figures S1a and S6 have been kept in the original form to exclude unnecessary gel lanes in between the relevant samples. Figure 2 below shows that the cropping was done in both cases in a way that does not alter the data. All original gel images can be found in the source data file.*

Figure 2. Comparison of the cropped gel images in Figures S1a (top panel) and S6 (bottom panel) with the original gel images. The areas of the gels selected for the figures have been highlighted.

2. Better provide the design schemes and staple sequences for the DNA origami designs for others to reproduce.

We have included the design of the 6HB, including DNA sequences, in the SI (Fig. S14).

3. Keep the unit for concentration of analytes consistent across the manuscript. For instance, for cTnI, use either ng/L or pmol/L. Plus, L (litter) should be spelled capital.

We thank the referee for bringing both the inconsistent use of units and the mistake in the capitalization of L to our attention.

Actions taken: *We have corrected the capitalization of L in the text and in the SI. We have revised the text to ensure that the molar and mass concentrations are used for cTnI in a consistent manner and reported both concentrations where relevant. We have added the following explanation for the choice of the units of concentration in the results section of the manuscript:*

“Despite the presence of troponin C, all serum troponin concentrations are reported as the concentration of cTnI. While molar concentrations make it easier to compare the sensitivity of detection between different analytes, cTnI levels are conventionally reported in literature as mass concentration (typically, as ng/mL or ng/L). We thus use both units of concentration in parallel, and the conversion between molar and mass concentration has been calculated using a molar mass of 23,900 g/mol for cTnI.”

Response to Referees

Title: *DNA Origami Signal Amplification in Lateral Flow Immunoassays*

We are happy about the positive reception of our work and the possibility of it being published in Nature Communications after the final revisions that we have carried out. We also greatly value the reviewers' recognition of the importance of our new results and their recommendation to highlight these findings in the main text.

After considering the options, we decided to keep the result figures in the SI. Differences in the style of sample preparation, strip design, data presentation, and the general breadth of experiments between cTnI and NfL prohibit the inclusion of the new results to the current logical structure and the flow of the figures. Regardless, we have revised the main text in the Introduction and Results sections to give the NfL results more visibility in the text.

Below you can find our detailed responses to all comments.

REVIEWERS' COMMENTS

Reviewer #2 (Remarks to the Author):

I am satisfied with the revisions undertaken by the authors.

However, it seems a bit of a shame to relegate the new results on NfL to the supplementary material. This sounds like a significant finding and the authors may wish to consider (possibly in discussion with the editor(s)) whether it would be possible to move it into the main text. Similarly, Table 1 in the response to referees might merit placement in the SI for easier access, and the normalized LoD values could be mentioned in the main text. In my opinion these are not essential revisions but may enhance the impact of the work.

As far as I am concerned the manuscript can now be accepted for publication by Nature Communications, with or without the minor changes suggested above.

Dr Katherine Dunn

We thank Dr. Dunn for the positive assessment of our work. After considering the options, we decided to keep the result figures concerning the NfL tests in the SI. Differences in the style of sample preparation, data presentation, and the general breadth of experiments between cTnI and NfL prohibit the inclusion of the new results to the current logical structure of the text and the flow of the figures.

Regardless, we fully agree that these are significant findings that deserve more space and we therefore added text in both the Introduction section (rows 74–77) and in the Results section (rows 269–273) to better emphasize and explain the results presented in the SI.

Additionally, while we agree with Dr. Dunn that Table 1 within the response letter enables easy comparison of the sensitivities of existing methods, we believe that the novelty of our concept should speak for itself. Moreover, different technologies offer different advantages and disadvantages depending on the context. For example, one method might be superior in terms of real life applicability and reproducibility, but fall short in terms of sensitivity to some other method. Thus we feel that any such table comparing a limited set of parameters entails the danger of simplification.

Reviewer #3 (Remarks to the Author):

The authors have addressed part of the suggestions as raised by the reviewer. It is highly recommended that those unaddressed comments being considered seriously to improve the current work to fulfill the high caliber requirement of Nat Commun.

Major concern

1. It remains unclear how the current test would perform on real clinical samples. LFIA tests have been available for decades, a methodology work on its own, without applications, is not sufficient to justify its publication on high profile journals like Nat Commun.

We appreciate the consistency of the reviewer, however, we already made the case in our previous response that a clinical study would go beyond the scope of this paper. However, we can assure that our method is currently being developed further, exactly for clinical and other large-scale studies.

Minor concerns

1. Results in the SI may be brought into main figures to make them more comprehensive and complete. For instance, the detection of NfL, detection of cTnl by commercial strips.

We thank the reviewer for acknowledging the value of the new experimental results. As we have already explained above, we decided against changing the figures and instead augmented the discussion on the new findings in the main text.

2. A spread sheet or table of staple strand sequences might be provided for the convenience of later users of this technology.

Together with the cadnano design map / strand diagram shown in Figure S14, we have now also included a table of the staple sequences in the SI (Table S1).